# COMMUNICATION-EFFICIENT HETEROGENEOUS FEDERATED LEARNING WITH GENERALIZED HEAVY-BALL MOMENTUM

## ABSTRACT

Federated Learning (FL) is the state-of-the-art approach for learning from decentralized data in privacy-constrained scenarios. As the current literature reports, the main problems associated with FL refer to system and statistical challenges: the former ones demand for efficient learning from edge devices, including lowering communication bandwidth and frequency, while the latter require algorithms robust to non-iidness. State-of-art approaches either guarantee convergence at increased communication cost or are not sufficiently robust to handle extreme heterogeneous local distributions. In this work we propose a novel generalization of the *heavy-ball* momentum, and present FEDHBM to effectively address statistical heterogeneity in FL without introducing any communication overhead. We conduct extensive experimentation on common FL vision and NLP datasets, showing that our FED-HBM algorithm empirically yields better model quality and higher convergence speed w.r.t. the state-of-art, especially in pathological non-iid scenarios. While being designed for cross-silo settings, we show how FEDHBM is applicable in moderate-to-high cross-device scenarios, and how good model initializations (e.g. pre-training) can be exploited for prompt acceleration. Extended experimentation on large-scale real-world federated datasets further corroborates the effectiveness of our approach for real-world FL applications. [1]

## 1 INTRODUCTION

The introduction of the Federated Learning (FL) paradigm in (McMahan et al., 2017) and FEDAVG algorithm has sparked a considerable interest in learning from decentralized data. In FL, a central server orchestrates an iterative two-step training process over several communication rounds consisting of: (i) local training on a potentially large number of clients, each having its own private data, and (ii) aggregation of the updated models into a shared global one. The intrinsic privacy-preserving nature of FL is appealing because it enables decentralized applications in cases where local data cannot be shared among clients. Yet, this very same characteristic of FL introduces also some challenges, because constraining the local optimization to use only the client's own data may cause statistical heterogeneity. This has been shown to hamper the convergence of FEDAVG (Hsu et al., 2019), increasing the number of communication rounds needed to reach a target model quality (Reddi et al., 2021) and the result at convergence. Recent advances in FL have tried to mitigate this problem, proposing new methods that possess strong theoretical guarantees even in the presence of a non-iid distribution of the local datasets but at the cost of increased communication. For instance, SCAFFOLD relies on additional control variables to correct the local client's updates, with experimentally better performances but with double the communication bandwidth requirements. Other recent algorithms (Karimireddy et al., 2021) require even more communication and also additional computation. Therefore, these solutions may be unsuitable in a regimen of limited communication resources, which is particularly relevant for applications with edge devices connected by slow, expensive and unreliable communication links (Kairouz et al., 2021). Moreover, albeit these methods are theoretically sound, in this paper we show experimental evidence that they are not sufficiently robust to handle cases of extreme heterogeneity (see fig. 1), confirming and extending what was found by Varno et al. (2022) for the specific case of FEDDYN (Acar et al., 2021).

---

[1]Code will be released upon acceptance

These considerations motivate the need for an FL algorithm that is both robust to client heterogeneity and communication-efficient by design. In this work, we try to answer the following research question:

*Is it possible to robustly speed-up federated optimization, even in extreme heterogeneous settings, without incurring in additional communication and computational costs?*

As a positive answer, we propose FEDHBM , a novel FL algorithm based on our generalization of the *heavy-ball* momentum (Polyak, 1964) to the federated setting. The underlying idea of FEDHBM is to exploit the models sent to a client at two subsequent rounds to calculate, locally on the client, a momentum term over a window of the last $\tau$ rounds of FedAvg. Intuitively, this formulation is equivalent to a moving average of velocity vectors and not gradients, thus providing a more direct and robust estimate of the global optimization trajectory that can be used as a client-drift correction. Our analysis reveals that the proposed momentum formulation has superior performance, being remarkably more stable in extreme heterogeneous scenarios. Additionally, by adding a local correction term, the presented method achieves faster convergence and improves the quality of the final model without any additional communication.

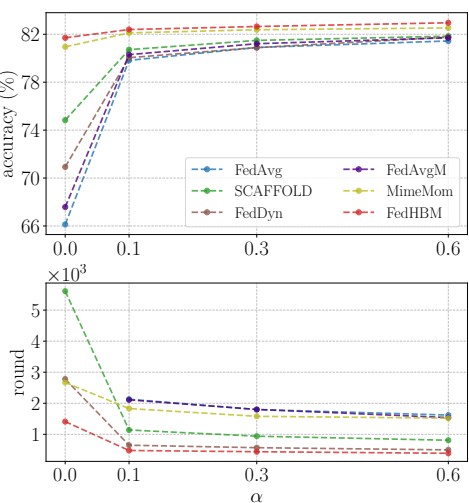

Figure 1: Impact of heterogeneity, from extreme ($\alpha = 0$) to moderate ($\alpha = 0.6$), on model quality (up) and convergence speed (no. of rounds to reach an absolute accuracy of 70%, bottom). Other algorithms are either robust to non-iidness (e.g. MIMEMOM ) or communication-efficient (e.g. FEDDYN ), while FEDHBM is the least affected.

**Contributions** We summarize the contributions of our work as follows:

- We shed a new light on the problem of communication-efficient FL under extreme statistical heterogeneity, and propose a framework based on a novel generalized heavy-ball formulation. We show that existing momentum-based FL algorithms can be regarderd as instances of this general framework and, within this same framework, we present FEDHBM , a robust and communication-efficient federated optimization algorithm.

- We perform an extensive empirical validation on common FL vision and NLP tasks, showing that FEDHBM yields both better model quality and higher convergence speed w.r.t. the state-of-art, especially in pathological non-iid scenarios. FEDHBM also shows remarkable flexibility with very low client participation rates, which makes it effective even in cross-device FL. In particular, we show how good model initializations, such as a pre-trained model, can be exploited to achieve a substantial acceleration.

- Extending the experimentation to large-scale real-world vision federated datasets, our analysis reveals robustness issues of even theoretically-proven algorithms. Conversely, these results corroborate the effectiveness of our approach for real-world FL applications.

**Related works** *The problem of non-iidness.* The detrimental effects of non-iid data have been first observed by (Zhao et al., 2018), who proposed to broadcast a small portion of public data to reduce the distance between local clients' distributions and partly recover the loss in performance. Alternatively, in (Li & Wang, 2019) the public data is kept server-side and used for knowledge distillation. However, such approaches require having data well suited for the purpose, which is a strong assumption. Having noticed that the performance loss comes from weight divergence, FedProx (Li et al., 2020) adds a regularization term to the loss function, penalizing the divergence from the global model. Nevertheless, in practical cases this was shown to be ineffective in addressing data heterogeneity Caldarola et al. (2022). Other works (Kopparapu & Lin, 2020; Zaccone et al., 2022; Zeng et al., 2022; Caldarola et al., 2021) have explored grouping clients based on their data distribution to alleviate the aggregation of divergent models.

*Stochastic Variance Reduction in FL.* Another research line applies stochastic variance reduction techniques in FL (Chen et al., 2021; Li et al., 2019). With SCAFFOLD, Karimireddy et al. (2020)

for the first time provided convergence guarantees in FL for arbitrarily heterogeneous data. The authors also shed light on the *client-drift* experienced in local optimization, which results in slow and unstable convergence. SCAFFOLD uses control variates to estimate the direction of the server model and clients' models: their difference is an estimate of the client drift and can be used to correct the local update. While well principled in theory and robust in practice, this approach requires double the communication to send the control variates back and forth. Similarly to (Karimireddy et al., 2020), we use a corrective term to alleviate the client drift during local optimization but our momentum term does not require any additional data exchange.

*ADMM and adaptivity.* Other methods are based on the Alternating Direction Method of Multipliers (Chen et al., 2022; Gong et al., 2022; Wang et al., 2022). In particular, FedDyn (Acar et al., 2021) dynamically modifies the loss function such that the model parameters converge to stationary points of the global empirical loss. Although technically it enjoys the same convergence properties of SCAFFOLD without suffering from its increased communication cost, in practical cases FedDyn has displayed problems in dealing with pathological non-iid settings (Varno et al., 2022).

*Momentum-based approaches.* Authors in (Hsu et al., 2019) found that using a server-side momentum effectively reduces the gap in accuracy between iid and non-iid scenarios. However, as highlighted in (Karimireddy et al., 2020), the source of slow and unstable convergence is the client drift experienced locally. FEDADC (Ozfatura et al., 2021) and FEDCM (Xu et al., 2021), albeit with slightly different but equivalent formulations, both propose to send the server momentum to clients to correct local updates. As a more general and theoretically proved framework, authors in (Karimireddy et al., 2021) proposed MIME to adapt an arbitrary centralized optimization algorithm to cross-device FL, by using a combination of control-variates and server optimizer state (e.g. momentum) at every client-update step. These statistics require an extra communication round and increased bandwidth, hence these algorithms are not communication-efficient. In this work we generalize the heavy-ball formulation by proposing a window wider than one round for momentum calculation. Within this framework, existing algorithms can be expressed as special case of our formulation. We investigate the role of a larger window, experimentally proving it is an enabling factor for dealing with extreme heterogeneity. We then propose our FEDHBM as a specific instantiation in which the window width being controlled by client participation leads to an algorithm robust and communication-efficient by design.

*Lowering communication requirements in FL.* Researchers have studied methods to reduce the memory needed for exchanging gradients in the distributed setting, for example by quantization (Alistarh et al., 2017) or by compression (Mishchenko et al., 2019; Koloskova* et al., 2020). In the context of FL, such ideas have been developed to meet the communication and scalability constraints (Reisizadeh et al., 2020), and to take into account non-iidness (Sattler et al., 2020). Our work focuses on the efficient use of the information already being sent in standard FedAvg, so additional techniques to compress that information remain orthogonal to our approach.

## 2   PROBLEM SETUP

**Notation.**   Throughout this work we adopt a unified notation both for ours and state-of-art algorithms, in a way compliant with the first work on FL (McMahan et al., 2017). We denote as $K \in \mathbb{N}^+$ the total number of clients who could participate in training, $C \in (0, 1]$ as the portion of them that participate in any round $t \in [T]$, and $\mathcal{S}$ and $\mathcal{S}^t$ as respectively the total set of clients and the set of clients participating in any round $t$. We indicate as $\mathcal{D}$ any data distribution, with $\mathcal{D}_i$ and $d_{i,j}$ respectively the local distribution and the $j$-th batch of size $B$ of client $i$, and $E$ as the number of local epochs. Conversely, $J_i := E\lceil |\mathcal{D}_i|/B \rceil$ is the number of local steps of client $i$, and $\eta, \eta_l$ indicate the global and local learning rates. In regards to the objective function, we call $f_\theta$ the function parameterized by model parameters $\theta$ and $L$ the loss function. More precisely, $\theta_{i,j}^t$ is the model of client $i$ at round $t$ before being presented with batch $j$, $\theta_{i,1}^t := \theta^{t-1}$ the model received by the server and $\theta_i^t := \theta_{i,J_i+1}^t$ the model trained by the $i$-th client and sent to the server for aggregation.

**Setting Cross-silo FL.**   In this setting, following the characterization in (Kairouz et al., 2021), the training nodes are expected to be different organizations or geo-distributed data centers. The number of such nodes is modest ($\mathcal{O}(10^2)$) and they are assumed to be almost always available and reliable. This makes it possible to maintain a state on nodes across two different rounds, and often the use of stateful clients is an indicator for an algorithm to be designed for this scenario. Usually, the problem

of FL in such a setting is cast as a finite-sum optimization problem, where each function is the local clients' loss function (eq. 1)

**Setting cross-device FL.** Differently from cross-silo FL, in the cross-device setting the clients are assumed to be possibly unreliable edge devices, with only a fraction of them available at any given time. As such, communication is the primary bottleneck. Most importantly, they can be massive in number ($\mathcal{O}(10^{10})$), so this motivates the fact that they should be stateless since each client is likely to participate only once in the training procedure. Following the characterization in (Karimireddy et al., 2021), being the number of clients enormous, this problem can be modeled by introducing the stochasticity client-level, over the possibly sampled clients (eq. 2).

CROSS-SILO:

$$\arg\min_{\theta \in \mathbb{R}^d} \sum_{k \in \mathcal{S}} \frac{|\mathcal{D}_k|}{|\mathcal{D}_{\mathcal{S}}|} \mathbb{E}_{(x,y) \sim \mathcal{D}_k}[L(f_\theta; (x,y))] \quad (1)$$

CROSS-DEVICE:

$$\arg\min_{\theta \in \mathbb{R}^d} \mathbb{E}_{i \sim S}\left[\sum_{i=1}^{|\mathcal{D}_i|} \frac{1}{|\mathcal{D}_i|} L(f_\theta; (x_i, y_i))\right] \quad (2)$$

**Cross-silo and cross-device in practice** The two aforementioned setups are however extreme cases, and real-world scenarios will likely enjoy some features from both settings. Previous FL works that address cross-silo FL usually experiment with a few hundred devices but account for low participation and unreliability, and treat communication as the primary bottleneck (Karimireddy et al., 2020; Acar et al., 2021). However, they are stateful, and this has raised concerns about their applicability in cross-device: in particular Karimireddy et al. (2021) noticed that the control variates in Karimireddy et al. (2020) get stale as clients are not seen again during training, and highlights that stateless clients reflect the different formulation in equations 2, 1. In this work we show that FEDHBM is robust to extremely low participation rates, and that it gets more effective as each client participates in the training process. Remarkably, our method succeeds in scenarios where even theoretically strong methods fail (see figure 2 and table 5).

## 3 METHOD

**Generalized heavy-ball momentum** The use of SGD with momentum is a common practice in deep learning (Krizhevsky et al., 2012; He et al., 2015) as it often provides faster convergence and better generalization (Yan et al., 2018). It consists in accumulating the directions of reduction of the objective function to stabilize the optimization dynamics. In this work, we propose a framework for using momentum in FL based on a novel generalization of Polyak's *heavy-ball* formulation (Polyak, 1964), as follows:

CLASSICAL HEAVY-BALL (POLYAK, 1964):

$$\theta^t \leftarrow \theta^{t-1} - \eta \nabla L(f_{\theta^{t-1}}) + \beta(\theta^{t-1} - \theta^{t-2}) \quad (3)$$

GENERALIZED HEAVY-BALL (GHB):

$$\theta^t \leftarrow \theta^{t-1} - \eta \nabla L(f_{\theta^{t-1}}) + \frac{\beta}{\tau}(\theta^{t-1} - \theta^{t-\tau-1}) \quad (4)$$

Namely, we propose to allow a wider $\tau$-window to be considered to estimate the momentum term. When setting $\tau = 1$, the above formulation falls back to SGD with Polyak's formulation, which is equivalent to a more common one that uses an additional variable to accumulate the previous directions (Liu et al., 2020; Sutskever et al., 2013). The main intuition behind our method is that the trajectory of the server updates over a window $\tau > 1$ provides a better estimate for the momentum term in a federated setting. This proves particularly important in FL because partial participation and non-iidness of local datasets tend to worsen the estimate of the global gradient. Intuitively, as $\tau$ increases, the momentum term increasingly incorporates information from a broader range of clients. A key observation is that when $\tau$ equals the average period length (e.g. $\tau = \frac{1}{C}$), under uniform client sampling, the momentum term contains the information on the global distribution and hence is optimal. We experimentally verified this hypothesis, demonstrating its validity in practice as we showed by purposely varying $\tau$ in figure 4.3. Within the GHB formulation, we also show that existing momentum-based FL algorithms implement the special case of GHB with $\tau = 1$, as it is shown in Table 1. However, in a FL scenario, implementing the GHB in eq. 4 for an arbitrary value of $\tau$ requires the server to send both models $\theta^{t-1}$ and $\theta^{t-\tau-1}$ to each client, resulting in a communication overhead of $1.5\times$ w.r.t. FedAvg. Namely, both methods in Xu et al. (2021); Ozfatura et al. (2021) incur in this overhead. To calculate such momentum in a communication efficient way, we can exploit the fact that a client participates multiple times in the training procedure, it has available the model

Table 1: Comparison of recent momentum-base FL algorithms within our generalized heavy-ball framework: FEDCM and FEDADC implement an equivalent update rule, since the only difference is a constant scaling on the gradient term (Liu et al., 2020). We generalize the momentum calculation over a window of $\tau$ rounds in GHB, recovering FEDCM (and FEDADC) when setting $\tau = 1$

| METHOD | UPDATE RULE IN ORIGINAL WORK | EQUIVALENT UPDATE RULE |
|---|---|---|
| FEDCM (Xu et al., 2021) | $\theta_{i,j+1}^t \leftarrow \theta_{i,j}^t - \eta_l(\alpha \nabla L(f_{\theta_{i,j}^t}, d_{i,j}) + (1-\alpha)m^{t-1})$ $m^t \leftarrow -\frac{1}{|\mathcal{S}^t|}\sum_{i=1}^{|\mathcal{S}^t|}(\theta_i^t - \theta^{t-1})$ $\theta^t \leftarrow \theta^{t-1} - \eta m^t$ | $\theta_{i,j+1}^t \leftarrow \theta_{i,j}^t - \eta_l(\nabla L(f_{\theta_{i,j}^t}, d_{i,j}) + \beta m^{t-1})$ $m^t \leftarrow \beta m^{t-1} + \frac{1}{|\mathcal{S}^t|}\sum_{i=1}^{|\mathcal{S}^t|}\sum_{j=1}^{J_i}\nabla L(f_{\theta_{i,j}^t}, d_{i,j})$ $\theta^t \leftarrow \theta^{t-1} - \frac{\eta}{|\mathcal{S}^t|}\sum_{i=1}^{|\mathcal{S}^t|}(\theta^t - \theta_i^t)$ |
| FEDADC (Ozfatura et al., 2021) | $\theta_{i,j+1}^t \leftarrow \theta_{i,j}^t - \eta_l(\nabla L(f_{\theta_{i,j}^t}, d_{i,j}) + 1/J_i m^{t-1})$ $m^t \leftarrow \frac{1}{\eta_l|\mathcal{S}^t|}\sum_{i=1}^{|\mathcal{S}^t|}(\theta^{t-1} - \theta_i^t) - (1-\beta)m^{t-1}$ $\theta^t \leftarrow \theta^{t-1} - \eta\eta_l m^t$ | $\theta_{i,j+1}^t \leftarrow \theta_{i,j}^t - \eta_l(\nabla L(f_{\theta_{i,j}^t}, d_{i,j}) + \beta m^{t-1})$ $m^t \leftarrow \beta m^{t-1} + \frac{1}{|\mathcal{S}^t|}\sum_{i=1}^{|\mathcal{S}^t|}\sum_{j=1}^{J_i}\nabla L(f_{\theta_{i,j}^t}, d_{i,j})$ $\theta^t \leftarrow \theta^{t-1} - \frac{\eta}{|\mathcal{S}^t|}\sum_{i=1}^{|\mathcal{S}^t|}(\theta^t - \theta_i^t)$ |
| MIMELITEMOM (Karimireddy et al., 2021) | $\theta_{i,j+1}^t \leftarrow \theta_{i,j}^t - \eta_l(\nabla L(f_{\theta_{i,j}^t}, d_{i,j}) + \beta m^{t-1})$ $m^t \leftarrow \beta m^{t-1} + \frac{1}{|\mathcal{S}^t|}\sum_{i=1}^{|\mathcal{S}^t|}\nabla L(f_{\theta^{t-1}}, \mathcal{D}_i)$ $\theta^t \leftarrow \theta^{t-1} - \frac{\eta}{|\mathcal{S}^t|}\sum_{i=1}^{|\mathcal{S}^t|}(\theta^t - \theta_i^t)$ | (NO EQUIVALENT) |
| GHB (ours) | $\theta_{i,j+1}^t \leftarrow \theta_{i,j}^t - \eta_l\nabla L(f_{\theta_{i,j}^t}, d_{i,j}) + \frac{\beta}{\tau J_i}(\theta^{t-1} - \theta^{t-\tau-1})$ $\theta^t \leftarrow \theta^{t-1} - \frac{\eta}{|\mathcal{S}^t|}\sum_{i=1}^{|\mathcal{S}^t|}(\theta^t - \theta_i^t)$ | $\theta_{i,j+1}^t \leftarrow \theta_{i,j}^t - \eta_l(\nabla L(f_{\theta_{i,j}^t}, d_{i,j}) + \beta m^{t-1})$ $m^t \leftarrow \frac{1}{\tau}\left(\sum_{t'=t-\tau+1}^{t}(\beta m^{t'-1} + \frac{1}{|\mathcal{S}^{t'}|}\sum_{i=1}^{|\mathcal{S}^{t'}|}\frac{1}{J_i}\sum_{j=1}^{J_i}\nabla L(f_{\theta_{i,j}^{t'}}, d_{i,j}))\right)$ $\theta^t \leftarrow \theta^{t-1} - \frac{\eta}{|\mathcal{S}^t|}\sum_{i=1}^{|\mathcal{S}^t|}(\theta^t - \theta_i^t)$ |

$\theta^{t-\tau_i-1}$ received at some round $t - \tau_i$. Hence, choosing $\tau = \tau_i$ does not involve additional data exchange. Let us remark that $\tau_i$ is not hand-tuned, but it is instead determined stochastically by client participation: in practice, under uniform sampling, on average each client automatically considers a window of length $\tau_i \approx 1/C$. In this sense, letting it be self-tuned resonates with the above intuition about considering the average period in which each client is sampled once. We show in section 4.3 that this choice is a good trade-off between required participation and performance. We name this communication efficient instance of our generalized momentum framework **LOCAL-GHB** (for a graphical intuition see fig. 5 in appendix).

**Algorithm 1:** FEDHBM and FedAvg

**Require:** initial model $\theta^0$, $K$ clients, $C$ participation ratio, $T$ number of total round, $B$ batch size, $\eta$ and $\eta_l$ learning rates.
1: **for** $t = 1$ to $T$ **do**
2:    $\mathcal{S}^t \leftarrow$ subset of clients $\sim \mathcal{U}(\mathcal{S}, \max(1, K \cdot C))$
3:    **for** $i \in \mathcal{S}^t$ **in parallel do**
4:      $\theta_{i,1}^t \leftarrow \theta^{t-1}$
5:      **for** $j = 1$ to $J_i$ **do**
6:        $m_{i,j}^t \leftarrow (\theta_{i,j}^t - \theta_i^{t-\tau_i})$ if $\theta_i^{t-\tau}$ is set **else 0**
7:        sample a mini-batch $d_{i,j}$ from $\mathcal{D}_i$
8:        $\theta_{i,j+1}^t \leftarrow \theta_{i,j}^t - \eta_l \nabla L(f_{\theta_{i,j}^t}, d_{i,j}) + \hat{\beta}_i m_{i,j}^t$
9:      **end for**
10:     save locally model $\theta_i^t$
11:    **end for**
12:    $\theta^t \leftarrow \theta^{t-1} - \eta \sum_{i \in \mathcal{S}^t}\frac{|\mathcal{D}_i|}{|\mathcal{D}_{\mathcal{S}^t}|}(\theta^{t-1} - \theta_i^t)$
13: **end for**

**FEDHBM** While a generalized momentum over a window $\tau > 1$ can better estimate the local correction to apply for embedding the updated information of other clients, the correction is not adjusted to the progressive drift of multiple local steps. To counteract this issue, we add a correction term specific to each client objective, such that it penalizes the direction of the last updates at round $t - \tau_i$ with respect to the progressive updates of local steps at current round $t$. This intuitions results in a slight modification of LOCAL-GHB, namely considering $\theta_{i,j}^t$ instead of $\theta^{t-1}$ and $\theta_i^{t-\tau_i}$ instead of $\theta^{t-\tau_i-1}$. As shown below, this results in an update rule consisting of two contributions, i) the same $\tau_i$-momentum of LOCAL-GHB and ii) a local correction term, penalizing the incremental updates of the current round with respect to the ones at round $t - \tau_i$.

We call **FEDHBM** the addition of such correction term to LOCAL-GHB. More formally, let us denote by $u_{i,j}^t$ the update performed by client $i$-th at step $j$-th for any round $t$, then FEDHBM update

rule can be written as follows:

$$\theta_{i,j+1}^t = \theta_{i,j}^t - \eta \nabla L(f_{\theta_{i,j}^t}, d_{i,j}) + \hat{\beta}_i(\theta_{i,j}^t - \theta_i^{t-\tau_i}) \tag{5}$$

$$= \theta_{i,j}^t - \eta \nabla L(f_{\theta_{i,j}^t}, d_{i,j}) + \hat{\beta}_i \left( \theta^{t-1} - \theta^{t-\tau_i-1} - \sum_{k=1}^{j} u_{i,k}^t + \sum_{k=1}^{J_i} u_{i,k}^{t-\tau_i} \right)$$

$$= \theta_{i,j}^t - \eta \nabla L(f_{\theta_{i,j}^t}, d_{i,j}) + \underbrace{\hat{\beta}_i(\theta^{t-1} - \theta^{t-\tau_i-1})}_{\text{LOCAL-GHB}} + \underbrace{\hat{\beta}_i \left( \sum_{k=1}^{J_i} u_{i,k}^{t-\tau_i} - \sum_{k=1}^{j} u_{i,k}^t \right)}_{\text{LOCAL CORRECTION}}$$

Let us notice that under uniform client sampling, it holds that $\mathbb{E}_{i \sim \mathcal{U}(\mathcal{S})}[\tau_i] \to \tau = \frac{1}{C}$. Consequently, the momentum factor in equation 5 is set as $\hat{\beta}_i := \frac{\beta C}{J_i}$

**Communication-efficiency and low participation**  Our efficient formulation relies on the fact that each client is selected more than once during training: this is reasonable in cross-silo settings, but may not hold in extreme cross-device scenarios. In this case, it would be still possible to leverage GHB , choosing the value of $\tau$ as a hyperparameter (with $1.5\times$ overhead). Even if in this work we focus on cases that are still tractable using our most efficient formulation 1, we show that large values of $\tau$ are still a robust choice in high cross-device settings: in particular, we show that is possible to consider a window starting from a common initialization and still recover the full acceleration obtained in cross-silo (see ablation study in section 4.3). Remarkably, robustness to very low participation rates is especially noticeable in our large-scale experiments in section 4.4, where other SOTA methods fail.

## 4 EXPERIMENTAL RESULTS

To validate our method, we run experiments on commonly used FL datasets across computer vision and NLP tasks. We then extend experimentation to large-scale real-world federated datasets. Comparing FEDHBM  with existing state-of-art algorithms, we find that: i) it is the most communication-efficient, ii) it yields the best model quality and iii) it provides a clear advantage in high cross-device scenarios, especially when starting training from pre-trained models.

### 4.1 SETUP

**Experimental protocol**  Our experimental baseline includes several state-of-the-art algorithms, including momentum-based methods (FEDAVGM Hsu et al. (2019), MIMEMOM and MIMELITEMOM (Karimireddy et al., 2021)). Since FEDCM and FEDADC correspond to our GHB with $\tau = 1$ and we report a full ablation on the value of $\tau$ (see section 4.3), they are not considered in the main result. All the results are reported in terms of mean top-1 accuracy over the last 100 rounds, averaged over 5 independent runs.

**Datasets and settings**  We consider image classification and next character/word prediction tasks. For the former, we use CIFAR-10/100, and for the latter SHAKESPEARE and STACKOVERFLOW . Following Hsu et al. (2020), for both CIFAR-10/100 we split the total datasets according to a Dirichlet distribution with concentration parameter $\alpha$, simulating two extreme levels of heterogeneity, corresponding to $\alpha = 0$ (NON-IID) and $\alpha = 10.000$ (IID). For SHAKESPEARE and STACKOVERFLOW we instead use the predefined splits. We consider two settings: the first one closer to cross-silo, we use CIFAR-10, CIFAR-100, and SHAKESPEARE , partitioning the datasets in $K = 100$ parts and choosing $C = 10\%$. The second is closer to cross-device: we choose $K = 500$ and $C = 1\%$ for both CIFAR's and use the natural split of STACKOVERFLOW dataset, corresponding to having $K = 40.000$ and $C = 0.12\%$. Let us remark that the level of non-iidness we introduce is extreme: in the non-iid cross-silo setting with CIFAR-100 each client only has samples belonging to a single class. Additional details about each setting are provided in table 3 of supplementary. We also present results on large-scale real-world FL vision datasets, LANDMARKS-USERS-160K and INATURALIST-USERS-120K , in section 4.4.

**Models**  Unless otherwise mentioned, for CIFAR-10/100 we use the version of LeNet-5 described in (Hsu et al., 2020), whereas for SHAKESPEARE and STACKOVERFLOW we use the same RNN and

Table 2: Number of rounds to reach a target accuracy w.r.t centralized of several SOTA FL algorithms with respect to ours ($\alpha \to 0$). In round brackets we report the speedup w.r.t FedAvg. Best result is in **bold**, second best is underlined.

| METHOD | COMM. OVERHEAD | CIFAR10 | | | | | |
| --- | --- | --- | --- | --- | --- | --- | --- |
| | | CROSS-SILO | | | CROSS-DEVICE | | |
| | | 70% | 80% | 90% | 70% | 80% | 90% |
| FEDAVG | 1x | 5520 (1.00x) | 9935 (1.00x) | - (-) | 5610 (1.00x) | - (-) | - (-) |
| FEDPROX | 1x | 5610 (0.98x) | 9935 (1.00x) | - (-) | 5610 (1.00x) | - (-) | - (-) |
| SCAFFOLD | 2x | 2800 (1.97x) | 5200 (1.91x) | - (-) | 2970 (1.89x) | 5270 (-) | - (-) |
| FEDDYN | 1x | 1000 (5.52x) | 1810 (5.49x) | - (-) | **1950 (2.88x)** | 3180 (-) | 7600 (-) |
| ADABEST | 1x | 5520 (1.00x) | 9935 (1.00x) | - (-) | 5610 (1.00x) | - (-) | - (-) |
| MIME | 2x | 3410 (1.62x) | 5180 (1.92x) | 9700 (-) | 3840 (1.46x) | 7340 (-) | - (-) |
| FEDAVGM | 1x | 5380 (1.03x) | 9500 (1.05x) | - (-) | 3480 (1.61x) | 5370 (-) | - (-) |
| FEDCM | 1.5x | 5400 (1.02x) | 9500 (1.05x) | - (-) | 3400 (1.65x) | 5300 (-) | - (-) |
| FEDADC | 1.5x | 5400 (1.02x) | 9500 (1.05x) | - (-) | 3400 (1.65x) | 5300 (-) | - (-) |
| MIMEMOM | 3x | 1500 (3.68x) | 2350 (4.23x) | 4450 (-) | 2490 (2.25x) | 3470 (-) | 7360 (-) |
| MIMELITEMOM | 2x | 2080 (2.65x) | 3320 (2.99x) | 6510 (-) | 3090 (1.82x) | 4510 (-) | 8490 (-) |
| **FEDHBM (ours)** | 1x | **770 (7.17x)** | **1270 (7.82x)** | **2560 (-)** | **1950 (2.88x)** | **2960 (-)** | **6510 (-)** |

LSTM used in Reddi et al. (2021); Karimireddy et al. (2021). Additional details about the datasets and the splits, the models' architecture, and the algorithms' hyperparameters are deferred to the appendix.

## 4.2 COMPARATIVE RESULTS

**Convergence speed** As it is possible to see from table 2, FEDHBM is consistently faster than the current state-of-the-art: it attains 70% of centralized accuracy with a speedup of $7.17\times$ and $2.88\times$ respectively in cross-silo and cross-device. Importantly, the reported results do not consider the additional slowdown introduced in MIME and SCAFFOLD due to increased communication: while usually being the second best, they require additional communication, which in practice nullifies the speed gains attained. Similar results hold also for CIFAR-100 and are reported in table 4 of supplementary. This evidence corroborates our claim that FEDHBM is the most communication-efficient method.

**Final model quality** As showed in tables Tables 3 and 4, FEDHBM consistently outperforms the other methods even when facing extreme non-iid clients' distributions, in both settings. FEDDYN improves FEDAVG on CIFAR-10, but fails to converge for CIFAR-100, in line with the results reported by Varno et al. (2022). Confirming the findings of Hsu et al. (2019), server-only momentum improves performance only in non-pathological scenarios, due to

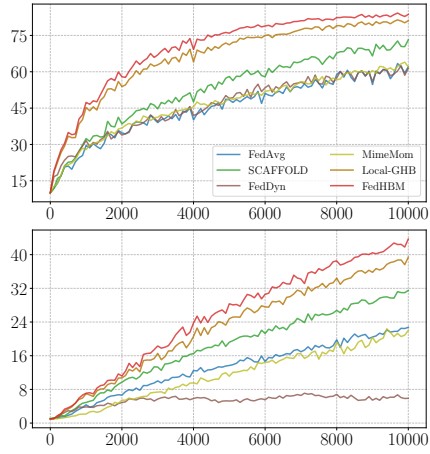

Figure 2: Comparing LOCAL-GHB and FEDHBM with other state-of-art approaches on CIFAR-10/100 (up and bottom respectively) using a ResNet-20, under extreme heterogeneity.

the client drift. Integrating the server momentum client side, MIMEMOM usually surpasses both SCAFFOLD and FEDDYN, especially in the presence of high heterogeneity. However these results are not consistent across architectures, since on our ResNet-20 experiments we found that MIMEMOM and FEDDYN fail to surpass FEDAVG (see figure 2). Conversely, FEDHBM consistently outperforms all the other algorithms across all settings, except for the extreme cross-device scenario of STACKOVERFLOW . This is mainly due to the fact that, since each client participates 1.5 times on average, FEDHBM most of the time cannot calculate its $\tau_i$-momentum. As we will show in Sec. 4.3, it is possible to easily circumvent this limitation, without introducing any communication overhead.

Table 3: Test accuracy (%) comparison of several SOTA FL algorithms on our **cross-silo** setting. Best result is in **bold**, second best is underlined.

| METHOD | CIFAR-10 | | CIFAR-100 | | SHAKESPEARE | |
|---|---|---|---|---|---|---|
| | NON-IID | IID | NON-IID | IID | NON-IID | IID |
| FEDAVG | $66.12 \pm 0.32$ | $83.11 \pm 0.34$ | $35.56 \pm 0.24$ | $49.74 \pm 0.22$ | $47.31 \pm 0.10$ | $47.08 \pm 0.17$ |
| FEDPROX | $66.12 \pm 0.32$ | $83.11 \pm 0.34$ | $35.48 \pm 0.30$ | $49.86 \pm 0.22$ | $47.30 \pm 0.10$ | $47.07 \pm 0.17$ |
| SCAFFOLD | $74.83 \pm 0.20$ | $82.93 \pm 0.25$ | $45.50 \pm 0.12$ | $49.41 \pm 0.40$ | $50.25 \pm 0.10$ | $50.13 \pm 0.10$ |
| FEDDYN | $70.93 \pm 0.18$ | $83.52 \pm 0.12$ | NaN | $\underline{51.95} \pm 0.17$ | $\underline{50.72} \pm 0.12$ | $\underline{50.80} \pm 0.16$ |
| ADABEST | $66.12 \pm 0.36$ | $83.11 \pm 0.38$ | $35.56 \pm 0.26$ | $49.74 \pm 0.25$ | $47.31 \pm 0.10$ | $47.08 \pm 0.17$ |
| MIME | $75.08 \pm 0.55$ | $83.13 \pm 0.46$ | $36.31 \pm 0.49$ | $50.87 \pm 0.36$ | $48.29 \pm 0.16$ | $48.49 \pm 0.15$ |
| FEDAVGM | $67.58 \pm 0.27$ | $\underline{83.60} \pm 0.31$ | $35.22 \pm 0.33$ | $50.68 \pm 0.25$ | $50.00 \pm 0.03$ | $50.41 \pm 0.08$ |
| FEDCM | $69.01 \pm 0.26$ | $83.39 \pm 0.30$ | $36.04 \pm 0.34$ | $50.18 \pm 0.50$ | $49.16 \pm 0.07$ | $50.45 \pm 0.09$ |
| FEDADC | $69.12 \pm 0.32$ | $83.41 \pm 0.32$ | $37.88 \pm 0.30$ | $50.16 \pm 0.41$ | $49.23 \pm 0.11$ | $50.42 \pm 0.12$ |
| MIMEMOM | $80.95 \pm 0.40$ | $83.11 \pm 0.20$ | $\underline{48.17} \pm 0.68$ | $50.60 \pm 0.11$ | $48.46 \pm 0.19$ | $48.89 \pm 0.25$ |
| MIMELITEMOM | $78.79 \pm 0.38$ | $83.23 \pm 0.29$ | $46.00 \pm 0.30$ | $50.66 \pm 0.10$ | $49.10 \pm 0.38$ | $49.39 \pm 0.32$ |
| **FEDHBM (ours)** | $\underline{81.71} \pm 0.15$ | $\mathbf{83.83} \pm 0.14$ | $\mathbf{50.41} \pm 0.51$ | $\mathbf{51.99} \pm 0.45$ | $\mathbf{51.33} \pm 0.08$ | $\mathbf{51.36} \pm 0.19$ |

Table 4: Test accuracy (%) comparison of several SOTA FL algorithms on our **cross-device** setting. Best result is in **bold**, second best is underlined.

| METHOD | CIFAR-10 | | CIFAR-100 | | STACKOVERFLOW |
|---|---|---|---|---|---|
| | NON-IID | IID | NON-IID | IID | NON-IID |
| FEDAVG | $66.08 \pm 0.15$ | $77.47 \pm 0.33$ | $35.31 \pm 0.31$ | $48.46 \pm 0.56$ | $24.02 \pm 0.41$ |
| FEDPROX | $65.92 \pm 0.26$ | $77.42 \pm 0.37$ | $35.32 \pm 0.20$ | $48.55 \pm 0.56$ | $23.88 \pm 0.42$ |
| SCAFFOLD | $74.20 \pm 0.12$ | $80.77 \pm 0.32$ | $\underline{44.59} \pm 0.38$ | $50.35 \pm 0.51$ | $\underline{24.77} \pm 0.41$ |
| FEDDYN | $\underline{77.79} \pm 0.73$ | $80.82 \pm 0.74$ | NaN | $50.46 \pm 0.31$ | $24.04 \pm 0.35$ |
| ADABEST | $65.91 \pm 0.25$ | $77.43 \pm 0.35$ | $35.31 \pm 0.31$ | $48.46 \pm 0.56$ | $24.01 \pm 0.4$ |
| MIME | $70.90 \pm 0.24$ | $77.64 \pm 0.17$ | $39.43 \pm 0.22$ | $48.30 \pm 0.20$ | $18.82 \pm 2.85$ |
| FEDAVGM | $73.90 \pm 0.97$ | $82.40 \pm 0.28$ | $38.11 \pm 1.04$ | $\underline{50.61} \pm 0.28$ | $24.07 \pm 0.35$ |
| FEDCM | $74.01 \pm 0.91$ | $81.36 \pm 0.25$ | $38.57 \pm 0.99$ | $50.56 \pm 0.38$ | $24.01 \pm 0.29$ |
| FEDADC | $73.96 \pm 0.89$ | $81.31 \pm 0.32$ | $38.52 \pm 1.01$ | $50.36 \pm 0.42$ | $23.96 \pm 0.23$ |
| MIMEMOM | $77.41 \pm 0.74$ | $\mathbf{82.87} \pm 0.22$ | $42.33 \pm 1.47$ | $50.12 \pm 0.29$ | $\mathbf{24.92} \pm 0.59$ |
| MIMELITEMOM | $76.41 \pm 1.15$ | $\underline{82.73} \pm 0.27$ | $41.23 \pm 2.57$ | $49.93 \pm 0.27$ | $23.30 \pm 3.46$ |
| **FEDHBM (ours)** | $\mathbf{79.31} \pm 0.45$ | $81.64 \pm 0.18$ | $\mathbf{48.69} \pm 0.95$ | $\mathbf{52.73} \pm 0.29$ | $24.47 \pm 0.40$ |

## 4.3 ABLATION STUDY

**The importance of $\tau$-window momentum in GHB** In figure 3 we show that the $\tau$-window momentum generalization introduced in our GHB formulation is crucial to effectively address extreme statistical heterogeneity. In fact constraining $\tau = 1$ fails at improving FEDAVG : this demonstrates that the correction provided by the momentum term is ineffective under extreme non-iidness when using the standard formulation. Both FEDCM and FEDADC are equivalent to GHB with $\tau = 1$ (cf. Table 1), hence they lead to the same results. Conversely, a wider window provides a steep enhancement both in convergence speed and final model quality, showing that our generalized momentum is the key factor for enabling excellent performance. Secondly our experiments show that our communication-efficient instance LOCAL-GHB , that allows each client

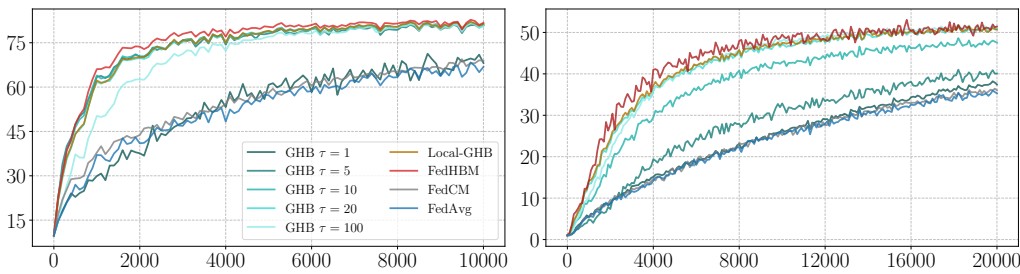

Figure 3: Ablation study on the size $\tau$ of the window for GHB on CIFAR-10/100 and comparison with LOCAL-GHB and FEDHBM , under extreme heterogeneity

to independently calculate its momentum term, reaches the same performance of GHB without the additional overhead of sending the global model of round $t - \tau$. Finally, thanks to the additional local correction term (eq. 1), FEDHBM always outperforms all the alternative solutions both in convergence speed and final model quality (see also figure 2).

**Addressing extreme cross-device scenarios**   Besides overall empirical success, we have shown that in the extreme cross-device of STACKOVERFLOW FEDHBM has diminished performance. This is due to the fact that most of the time the momentum term will be equal to zero (line 6 in algorithm 1). To address such limitation, we propose just to use the simpler formulation of LOCAL-GHB the first time each client is selected, using as past model the initial server model. From the second time on, each client uses the formulation in eq. 5. Let us note that this does not require additional communication: when training a model from scratch, it is necessary to only know the initialization algorithm and the seed for the random number generator to recover the very same model client side. We denote this variation FedHBM-shared. To further investigate the impact of initial model selection, we conducted experiments in which clients were allowed to choose a distinct random initialization, referred to as FedHBM-random. As it is possible to see from figure 4a, both solutions make our algorithm to recover its full performance gains, underscoring the resilience of our approach.

**Use of pre-trained models**   Following the practice highlighted above, when training from a pre-trained model it is possible to use it as past model for all clients. The availability of the pre-trained model does not constitute a communication-hampering factor, since it can be asynchronously downloaded from a server different than the FL training orchestrator. We experiment by letting the initial server model have the feature extractor initialized from a pre-trained model (on CIFAR-100 for CIFAR-10 and vice versa). As illustrated in figures 4b and 4c, this modification allows regaining full speed from early rounds of training, thereby demonstrating the efficacy of leveraging a well-initialized model for prompt acceleration.

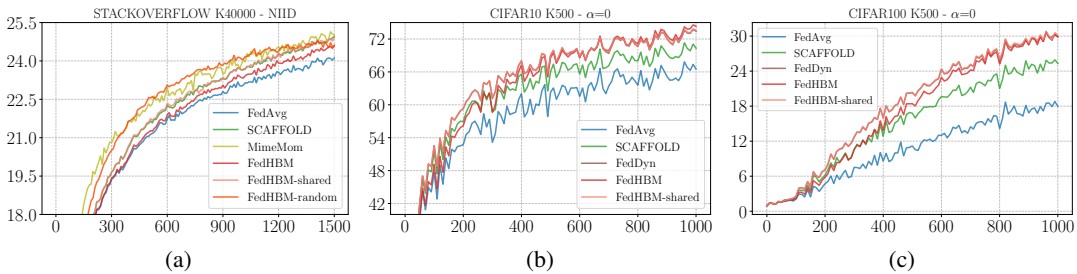

Figure 4: Effect of using a shared model as initialization. For CIFAR's we show the impact of using a pre-trained backbone, while for STACKOVERFLOW we analyze the use of a random shared or independent model initialization.

### 4.4   RESULTS IN LARGE-SCALE REAL-WORLD SCENARIOS

To further corroborate the results presented in controlled scenarios on common FL datasets, in this section we extended our experimentation to real-world large-scale FL vision datasets, following Hsu et al. (2020). Results in table 5 show that FEDHBM outperforms SOTA methods even in large-scale applications. Importantly, it shows superior robustness, as we show failure cases of even theoretically-backed algorithms (e.g. SCAFFOLD ), despite careful and broad hyperparameter search. In particular, for MIMEMOM we leveraged the official JAX implementation provided by authors (see section B.1 for details).

## 5   CONCLUSIONS

We introduced a framework based on a novel generalized *heavy-ball* momentum (GHB ) formulation for FL; in particular, we showed that existing momentum-based FL algorithms are instances of this general framework. Within it we proposed FEDHBM , which outperforms the state-of-the-art

Table 5: Test accuracy (%) comparison of best SOTA FL algorithms on LANDMARKS-USERS-160K and INATURALIST-USERS-120K

| METHOD | COMM. OVERHEAD | LANDMARKS-USERS-160K | INATURALIST-USERS-120K | | |
| --- | --- | --- | --- | --- | --- |
| | | $C \approx 0.79\%$ | $C \approx 0.1\%$ | $C \approx 0.5\%$ | $C \approx 1\%$ |
| FEDAVG | $1\times$ | $60.31 \pm 0.18$ | $38.03 \pm 0.84$ | $45.25 \pm 0.07$ | $47.59 \pm 0.13$ |
| SCAFFOLD | $2\times$ | $61.03 \pm 0.08$ | $0.0$ | $0.0$ | $0.0$ |
| FEDAVGM | $1\times$ | $61.50 \pm 0.22$ | $41.34 \pm 0.38$ | $46.08 \pm 0.09$ | $48.37 \pm 0.07$ |
| MIMEMOM | $3\times$ | $0.0$ | $0.0$ | $0.0$ | $0.0$ |
| **FEDHBM (ours)** | $1\times$ | $\mathbf{65.41 \pm 0.17}$ | $\mathbf{41.64 \pm 0.18}$ | $\mathbf{47.33 \pm 0.04}$ | $\mathbf{49.80 \pm 0.05}$ |

approaches in terms of both model quality and convergence speed. Remarkably, we showed that FEDHBM is the most robust to statistical heterogeneity and performs favorably even in high cross-device settings and real-world scenarios. The generality and versatility of the novel generalized heavy-ball momentum formulation we propose expands its potential applications to a wider range of scenarios where communication is a bottleneck, such as distributed learning.

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

# A    Visualization of $\tau_i$-windows in FedHBM

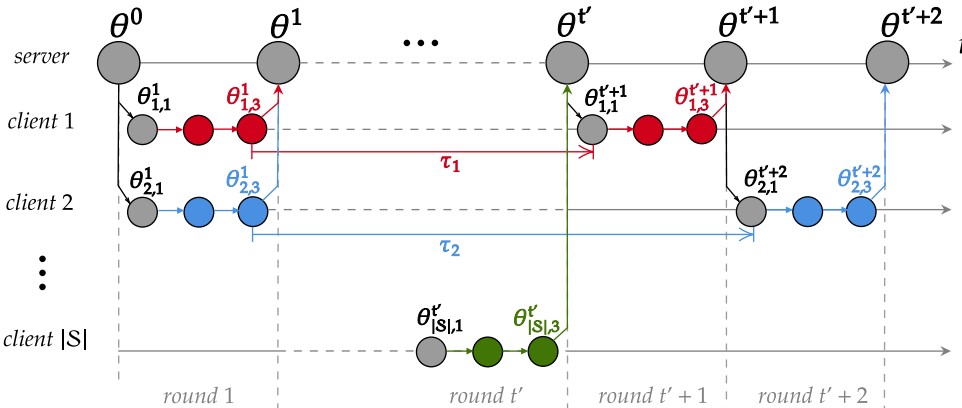

Figure 5: Illustration of momentum term calculation in FedHBM , considering different $\tau_i$ windows for different clients. On the horizontal axis are shown the rounds, while on the vertical axis we depict the clients and the server.

Figure 5 depicts the server-client interactions in standard FL training. First, a global server model $\theta^{t-1}$ is sent to all selected clients in round $t$, then this model is used as starting point $\theta^t_{i,j}$ for local training, where $i$ indicates the client index and $j$ is the local training step. Finally, at the end of round $t$, the model trained by each client, denoted as $\theta^t_i$, is sent to the server for aggregation. The number of rounds between two subsequent times a client $i$ is selected is called $\tau_i$. Different clients can have different $\tau_i$, i.e., *"windows"*, as depicted in the figure for clients $1$ and $2$. Moreover, the number of rounds $\tau_i$ is not fixed for a client, as this may be sampled with varying frequency. However, under uniform client sampling assumption, we expect to have $\tau_i \to \tau = 1/C$.

# B    Experiments

## B.1    Dataset and models

**Cifar-10/100**    We consider Cifar-10 and Cifar-100 to experiment with image classification tasks, each one respectively having 10 and 100 classes. For all methods, training images are pre-processed by applying random crops, followed by random horizontal flips. Both training and test images are finally normalized according to their mean and standard deviation. As the main model for experimentation, we used a model similar to LeNet-5 as proposed in (Hsu et al., 2020). To further validate our findings, we also employed a ResNet-20 as described in (He et al., 2015), following the implementation provided in (Idelbayev). Since batch normalization Ioffe & Szegedy (2015) layers have been shown to hamper performance in learning from decentralized data with skewed label distribution (Hsieh et al., 2020), we replaced them with group normalization (Wu & He, 2018), using two groups in each layer. For a fair comparison, we used the same modified network also in centralized training. We report the result of centralized training for reference in Table 6: as per the hyperparameters, we use $64$ for the batch size, $0.01$ and $0.1$ for the learning rate respectively for the LeNet and the ResNet-20 and $0.9$ for momentum. We trained both models on both datasets for $150$ epochs using a cosine annealing learning rate scheduler.

To simulate FL training on those datasets, for both our cross-silo and cross-device settings we split the total datasets according to a Dirichlet distribution with concentration parameter $\alpha$, following Hsu et al. (2020).

In practice, we draw a multinomial $q_i \sim \mathbf{Dir}(\alpha p)$ from a Dirichlet distribution, where $p$ describes a prior class distribution over $N$ classes, and $\alpha$ controls the iidness among all clients: the greater $\alpha$ the more homogeneous the clients data distributions will be. After drawing the class distributions $q_i$, for every client $i$, we sample training examples for each class according to $q_i$ without replacement. In

the main paper, we considered only two levels of heterogeneity: the former corresponds to setting $\alpha = 0$ and is used to simulate a pathological non-iid scenario, while the latter uses $\alpha = 10k$, and corresponds to having homogeneous local datasets. To further investigate the impact of heterogeneity, we provide the results for different values of $\alpha$ in section B.4 of this supplementary.

**Shakespeare** The Shakespeare language modeling dataset is created by collating the collective works of William Shakespeare and originally comprises 715 clients, with each client denoting a speaking role. However, for this study, a different approach was used, adopting the LEAF (Caldas et al., 2019) framework to split the dataset among 100 devices and restrict the number of data points per device to 2000. The non-IID dataset is formed by assigning each device to a specific role, and the local dataset for each device contains the sentences from that role. Conversely, the IID dataset is created by randomly distributing sentences from all roles across the devices.

For this task, we have employed a two-layer Long Short-Term Memory (LSTM) classifier, consisting of 100 hidden units and an 8-dimensional embedding layer. Our objective is to predict the next character in a sequence, where there are a total of 80 possible character classes. The model takes in a sequence of 80 characters as input, and for each character, it learns an 8-dimensional representation. The final output of the model is a single character prediction for each training example, achieved through the use of 2 LSTM layers and a densely-connected layer followed by a softmax. This model architecture is the same used by (Li et al., 2020; Acar et al., 2021).

We report the result of centralized training for reference in Table 6: we train for 75 epochs with constant learning rate, using as hyperparameters 100 for the batch size, 1 for the learning rate, 0.0001 for the weight decay and no momentum.

**StackOverflow** The Stack Overflow dataset is a language modeling corpus that comprises questions and answers from the popular Q&A website, StackOverflow. Initially, the dataset consists of 342477 unique users but for, practical reasons, we limit our analysis to a subset of $40k$ users. Our goal is to perform the next-word prediction on these text sequences. To achieve this, we utilize a Recurrent Neural Network (RNN) that first learns a 96-dimensional representation for each word in a sentence and then processes them through a single LSTM layer with a hidden dimension of 670. Finally, the model generates predictions using a densely connected softmax output layer. The model and the preprocessing steps are the same as in (Reddi et al., 2021).

We report the result of centralized training for reference in Table 6: as per the hyperparameters, we use 16 for the batch size, $10^{-1/2}$ for the learning rate and no momentum or weight decay. We train for 50 epochs with a constant learning rate.

Given the size of the test dataset, testing on STACKOVERFLOW is conducted on a subset of them made by 10000 randomly chosen test examples, selected at the beginning of training.

**Large-scale real-world datasets** As large-scale real-world datasets for our experimentation we follow Hsu et al. (2020). LANDMARKS-USERS-160K is composed of $\approx 164k$ images belonging to $\approx 2000$ classes, realistically split among 1262 clients. INATURALIST-USERS-120K is composed of $\approx 120k$ images belonging to $\approx 1200$ classes, split among 9275 clients. For both datasets we use a MobileNetV2 pretrained on ImageNet. These datasets are challenging to train not only because of their inherent complexity (size of images, number of classes), but also because usually at each round a very small portion of clients is selected. In particular, for LANDMARKS-USERS-160K we sample 10 clients per round, while for INATURALIST-USERS-120K we experiment different participation rates, sampling 10, 50 or 100 clients per round. In the main paper we choose to report the participation rate instead of the number of sampled clients to better highlight that the tested scenarios are closer to a cross-device setting, which is the most challenging for algorithms based on client participation, like SCAFFOLD and ours. As per the model, for both datasets we use a MobileNetV2 pretrained on ImageNet.

**Hyper-parameters** For ease of consultation, we report the hyper-parameters grids as well as the chosen values in Table 7. For LANDMARKS-USERS-160K and INATURALIST-USERS-120K we only test the best SOTA algorithms, that is FEDAVG and FEDAVGM as baselines, SCAFFOLD and MIMEMOM . For all algorithms we perform $E = 5$ local epochs, and searched $\eta \in \{0.1, 1\}$ and $\eta_l \in \{0.01, 0.1\}$, and found $\eta = 0.1, \eta_l = 0.1$ works best for FEDAVGM , while $\eta = 1, \eta_l = 0.1$ works

best for the others. For INATURALIST-USERS-120K , we had to enlarge the grid for SCAFFOLD and MIMEMOM : for both we searched $\eta \in \{10^{-1}, 10^{-1/2}, 1\}$ and $\eta_l \in \{10^{-2}, 10^{-3/2}, 10^{-1}\}$. Despite our effort, we did not obtain positive results for these algorithms. Failure cases of these theoretically-backed algorithms are interesting to analyze, but are out of the scope of this work. For SCAFFOLD , most likely it is because of very low participation rates, which causes control variates to become stale, as highlighted in Karimireddy et al. (2021).

For simplicity, in all our FL experiments we did not use any learning rate scheduler. In fact, while using strategies to change the learning rate as training proceeds is in general beneficial, this would result in a difficult tuning of hyper-parameters associated with the scheduler, since the algorithms present very different convergence rates. As such, our result must be read as a comparison between different algorithms, and the results of centralized training in Table 6 are intended just as a reference.

Table 6: Test accuracy (%) of centralized training over datasets and models used. Results are reported in term of mean top-1 accuracy over the last 10 epochs, averaged over 5 independent runs.

| DATASET | ACC. CENTRALIZED (%) |
|---|---|
| CIFAR-10 W/ LENET | $86.48 \pm 0.22$ |
| CIFAR-10 W/ RESNET-20 | $89.05 \pm 0.44$ |
| CIFAR-100 W/ LENET | $57.00 \pm 0.09$ |
| CIFAR-100 W/ RESNET-20 | $62.21 \pm 0.85$ |
| SHAKESPEARE | $52.00 \pm 0.16$ |
| STACKOVERFLOW | $28.50 \pm 0.25$ |

Table 7: Hyper-parameter search grid for each combination of method and dataset. The best performing values are indicated in **bold**.

| METHOD | HPARAM | CIFAR-10/100 | | SHAKESPEARE | STACKOVERFLOW |
|---|---|---|---|---|---|
| | | LENET | RESNET-20 | | |
| ALL FL | wd | [**0.001**, 0.0008, 0.0004] | [0.0001, **0.00001**] | [**0**, 0.0001, 0.00001] | [**0**, 0.0001, 0.00001] |
| | $B$ | 64 | 64 | 100 | 16 |
| FEDAVG | $\eta$ | [**1**, 0.5, 0.1] | [**1**, 0.1] | [**1**, 0.5, 0.1] | [**1**, 0.5, 0.1] |
| | $\eta_l$ | [0.1, 0.05, **0.01**] | [**0.1**, 0.01] | [**1**, 0.5, 0.1] | [1, 0.5, **0.3**, 0.1] |
| FEDPROX | $\eta$ | [**1**, 0.5, 0.1] | - | [**1**, 0.5, 0.1] | [**1**, 0.5, 0.1] |
| | $\eta_l$ | [0.1, 0.05, **0.01**] | - | [**1**, 0.5, 0.1] | [1, 0.5, **0.3**, 0.1] |
| | $\mu$ | [**0.1**, 0.01, 0.001] | - | [0.1, 0.01, 0.001, **0.0001**] | [0.1, **0.01**, 0.001, 0.0001] |
| SCAFFOLD | $\eta$ | [**1**, 0.5, 0.1] | [**1**, 0.1] | [**1**, 0.5, 0.1] | [**1**, 0.5, 0.1] |
| | $\eta_l$ | [0.1, 0.05, **0.01**] | [**0.1**, 0.01] | [**1**, 0.5, 0.1] | [1, 0.5, **0.3**, 0.1] |
| FEDDYN | $\eta$ | [**1**, 0.5, 0.1] | [**1**, 0.1] | [**1**, 0.5, 0.1] | [**1**, 0.5, 0.1] |
| | $\eta_l$ | [0.1, 0.05, **0.01**] | [0.1, **0.01**] | [**1**, 0.5, 0.1] | [1, 0.5, **0.3**, 0.1] |
| | $\alpha$ | [0.1, 0.01, **0.001**] | [0.1, 0.01, **0.001**] | [0.1, **0.009**, 0.001] | [**0.1**, 0.009, 0.001] |
| ADABEST | $\eta$ | [**1**, 0.5, 0.1] | - | [**1**, 0.5, 0.1] | [**1**, 0.5, 0.1] |
| | $\eta_l$ | [0.1, 0.05, **0.01**] | - | [**1**, 0.5, 0.1] | [1, 0.5, **0.3**, 0.1] |
| | $\alpha$ | [0.1, 0.01, **0.001**] | - | [0.1, **0.009**, 0.001] | [**0.1**, 0.009, 0.001] |
| MIME | $\eta$ | [**1**, 0.5, 0.1] | [**1**, 0.1] | [**1**, 0.5, 0.1] | [**1**, 0.5, 0.1] |
| | $\eta_l$ | [0.1, 0.05, **0.01**] | [0.1, **0.01**] | [**1**, 0.5, 0.1] | [1, 0.5, **0.3**, 0.1] |
| FEDAVGM | $\eta$ | [1, 0.5, **0.1**] | [1, **0.1**] | [1, 0.5, **0.1**] | [**1**, 0.5, 0.1] |
| | $\eta_l$ | [0.1, 0.05, **0.01**] | [**0.1**, 0.01] | [**1**, 0.5, 0.1] | [1, 0.5, **0.3**, 0.1] |
| | $\beta$ | [0.99, **0.9**] | [0.99, **0.9**] | [0.99, **0.9**] | [0.99, **0.9**] |
| MIMEMOM | $\eta$ | [1, 0.5, **0.1**] | [1, **0.1**] | [1, 0.5, **0.1**] | [**1**, 0.5, 0.1] |
| | $\eta_l$ | [0.1, 0.05, **0.01**] | [0.1, **0.01**] | [**1**, 0.5, 0.1] | [1, 0.5, 0.3, **0.1**] |
| | $\beta$ | [0.99, **0.9**] | [0.99, **0.9**] | [0.99, **0.9**] | [0.99, **0.9**] |
| MIMELITEMOM | $\eta$ | [1, 0.5, **0.1**] | [1, **0.1**] | [1, 0.5, **0.1**] | [**1**, 0.5, 0.1] |
| | $\eta_l$ | [0.1, 0.05, **0.01**] | [0.1, **0.01**] | [**1**, 0.5, 0.1] | [1, 0.5, 0.3, **0.1**] |
| | $\beta$ | [0.99, **0.9**] | [0.99, **0.9**] | [0.99, **0.9**] | [0.99, **0.9**] |
| FEDCM | $\eta$ | [1, 0.5, **0.1**] | - | [1, 0.5, **0.1**] | - |
| | $\eta_l$ | [1, 0.5, **0.1**] | - | [**1**, 0.5, 0.1] | - |
| | $\alpha$ | [0.05, **0.1**, 0.5] | - | [0.05, **0.1**, 0.5] | - |
| **FEDHBM (ours)** | $\eta$ | [**1**, 0.5, 0.1] | [**1**, 0.1] | [**1**, 0.5, 0.1] | [**1**, 0.5, 0.1] |
| | $\eta_l$ | [0.1, 0.05, **0.01**] | [0.1, **0.01**] | [**1**, 0.5, 0.1] | [1, 0.5, **0.3**, 0.1] |
| | $\beta$ | [**1**, 0.99, 0.9] | [**1**, 0.99, 0.9] | [**1**, 0.99, 0.9] | [**1**, 0.99, 0.9] |

## B.2 SIMULATING CROSS-SILO AND CROSS-DEVICE

In the main paper, we provide experimental results on two different settings, devised to simulate the characteristics of cross-silo and cross-device FL as described in (Kairouz et al., 2021). For both settings, we test two different tasks: image classification and NLP. Table 8 illustrates our use of dataset splits to simulate the two settings.

**Cross-silo FL**  As per image classification, we use CIFAR-10-100, splitting the dataset among 100 clients and selecting at each round 10% of them. This leads to each device being selected a relatively high number of times, and so represents the most suitable setting for stateful algorithms, like ours. For the next-character prediction task, we use LEAF (Caldas et al., 2019) to split the text dataset of SHAKESPEARE dialogues among 100 clients and select 10% of them at each round.

**Cross-device FL**  As per image classification, we use a different split of CIFAR-10-100, diving the dataset among 500 clients and selecting at each round 1% of them. In this scenario, each client is selected 10 times fewer than the cross-silo, and, considering the difficulty of the task, we found this to be reasonably challenging as a cross-device setting. For the next-word prediction task, our split of STACKOVERFLOW dataset consists of $40k$ local datasets assigned to clients, and at each round, we select 50 of them. This division represents the most challenging scenario for stateful algorithms, since each client is selected on average every 800 rounds, and this usually leads to stale local information. In the main paper, we showed that even in this case FEDHBM effectively addresses this kind of scenario, since the local information is used to calculate a momentum term.

Table 8: Details about datasets' split used for our experiments

|  | CIFAR-10 | | CIFAR-100 | | SHAKESPEARE | STACKOVERFLOW |
|---|---|---|---|---|---|---|
|  | CROSS-SILO | CROSS-DEVICE | CROSS-SILO | CROSS-DEVICE | | |
| Clients | 100 | 500 | 100 | 500 | 100 | 40k |
| Avg. examples per client | 500 | 100 | 500 | 100 | 2000 | 428 |
| Number of local steps | 8 | 2 | 8 | 2 | 20 | 27 |
| Number of clients per round | 10 | 5 | 10 | 5 | 10 | 50 |
| Total number of rounds | 10k | 10k | 20k | 20k | 250 | 1500 |
| Average participation (round) | 1k | 100 | 2k | 200 | 25 | 1.5 |

**Implementation details and practicality of experiments**  We implemented all the tested algorithms and training procedures in a single codebase, using PYTORCH 1.10 framework, compiled with CUDA 10.2. The federated learning setup is simulated by using a single node equipped with 11 Intel(R) Core(TM) i7-6850K CPUs and 4 NVIDIA GeForce GTX 1070 GPUs, running in a sequential manner (on a single GPU) the parallel client training and the following aggregation by the central server. Under these conditions, a single FedAvg experiment on CIFAR-10/100 using LeNet-5 takes $\approx 6$ (LeNet) and $\approx 7$ hours (ResNet-20) to run $10k$ rounds in our cross-silo scenario. For SCAFFOLD we use the `"option II"` of their algorithm (Karimireddy et al., 2020) to calculate the client controls, incurring almost no overhead in our simulations. Conversely, all MIME's methods incur a significant overhead due to the additional round needed to calculate the full-batch gradients, taking $\approx 12$ hours. On SHAKESPEARE and STACKOVERFLOW , FedAvg takes $\approx 22$ minutes and $\approx 3.5$ hours to run respectively 250 and 1500 rounds.

### B.3  ADDITIONAL EXPERIMENTS

In this section, we provide additional experiments to the ones reported in the main paper. Namely, we present the complete results about convergence speed and show additional numerical results for the experiments in cross-silo using a ResNet-20, as partially presented in Figure 2 of the main paper.

**Convergence speed on CIFAR-100**  Table 9 completes the results presented in Table 2 of the main paper, reporting the results on CIFAR-100. As it is shown, FEDHBM is the fastest algorithm for all the target values of accuracy reported, in both cross-silo and cross-device settings. FEDHBM is $1.76\times$ and $1.67\times$ faster than MIMEMOM in reaching 60% of target accuracy, respectively in cross-silo and cross-device settings. Let us note that the reported results do not account for the communication overhead introduced by other methods, hence the showed gains are lower bounds for the actual speedups we expect in real scenarios.

**Results on ResNet-20**  Table 10 reports the final model quality on our **cross-silo** setting for CIFAR-10/100 using the ResNet-20. These results confirm the ones reported in the main paper, and prove generalization to a more complex model architecture on image classification. In particular, SCAF-FOLD (Karimireddy et al., 2020) confirms its effectiveness in handling heterogeneity, improving

Table 9: Number of rounds to reach a target accuracy w.r.t centralized of several SOTA FL algorithms with respect to ours ($\alpha = 0$) for CIFAR100. In round brackets we report the speedup w.r.t FedAvg. Best result is reported in **bold**, second best is underlined.

| METHOD | COMM. OVERHEAD | CIFAR100 | | | | | |
|---|---|---|---|---|---|---|---|
| | | CROSS-SILO | | | CROSS-DEVICE | | |
| | | 60% | 70% | 80% | 60% | 70% | 80% |
| FEDAVG | 1x | 15540 (1.00x) | - (-) | - (-) | 16270 (1.00x) | - (-) | - (-) |
| FEDPROX | 1x | 16050 (0.97x) | - (-) | - (-) | 16330 (1.00x) | - (-) | - (-) |
| SCAFFOLD | 2x | 7290 (2.13x) | 10580 (-) | 16410 (-) | 8600 (1.89x) | 11870 (-) | 18220 (-) |
| FEDDYN | 1x | - (-) | - (-) | - (-) | - (-) | - (-) | - (-) |
| ADABEST | 1x | 15540 (1.00x) | - (-) | - (-) | 16270 (1.00x) | - (-) | - (-) |
| MIME | 2x | 15370 (1.01x) | 19810 (-) | - (-) | 11870 (1.37x) | 17070 (-) | - (-) |
| FEDAVGM | 1x | 16510 (0.94x) | - (-) | - (-) | 12740 (1.28x) | 19370 (-) | - (-) |
| MIMEMOM | 3x | 4530 (3.43x) | 6100 (-) | 8810 (-) | 8540 (1.91x) | 12370 (-) | - (-) |
| MIMELITEMOM | 2x | 6360 (2.44x) | 8810 (-) | 13630 (-) | 10130 (1.61x) | 14320 (-) | - (-) |
| **FEDHBM (ours)** | 1x | **2570 (6.05x)** | **3580 (-)** | **5930 (-)** | **5090 (3.20x)** | **7460 (-)** | **11870 (-)** |

Table 10: Test accuracy (%) comparison of several SOTA FL algorithms on our **cross-silo** setting, using a ResNet-20. Best result is in **bold**, second best is underlined.

| METHOD | NON-IID | | IID | |
|---|---|---|---|---|
| | CIFAR-10 | CIFAR-100 | CIFAR-10 | CIFAR-100 |
| FEDAVG | $61.03 \pm 1.06$ | $21.94 \pm 0.88$ | $86.54 \pm 0.20$ | $58.57 \pm 0.40$ |
| SCAFFOLD | $71.78 \pm 1.67$ | $30.73 \pm 1.31$ | $86.82 \pm 0.31$ | $58.05 \pm 0.66$ |
| FEDDYN | $60.25 \pm 3.05$ | $6.02 \pm 0.52$ | $86.96 \pm 0.33$ | $60.76 \pm 0.71$ |
| MIME | $53.69 \pm 2.89$ | $9.00 \pm 0.41$ | $86.66 \pm 0.15$ | $59.00 \pm 0.32$ |
| FEDAVGM | $65.99 \pm 2.24$ | $22.81 \pm 0.80$ | $87.74 \pm 0.27$ | $58.68 \pm 0.89$ |
| MIMEMOM | $69.25 \pm 3.64$ | $21.67 \pm 1.11$ | $88.01 \pm 0.14$ | $60.46 \pm 0.63$ |
| MIMELITEMOM | $57.03 \pm 0.91$ | $14.39 \pm 0.58$ | $87.98 \pm 0.39$ | $59.22 \pm 0.47$ |
| LOCAL-GHB (ours) | $80.56 \pm 0.28$ | $38.17 \pm 1.02$ | $88.76 \pm 0.15$ | $62.00 \pm 0.49$ |
| **FEDHBM (ours)** | **83.39** $\pm 0.34$ | **42.46** $\pm 0.76$ | **89.23** $\pm 0.15$ | **62.48** $\pm 0.48$ |

FEDAVG by approximately 10 accuracy points on both CIFAR-10/100. Instead FEDDYN does not bring any improvement w.r.t. FEDAVG on CIFAR-10 and it actually performs significantly worse on CIFAR-100 in the cross-silo setting. Concerning the family of MIME algorithms, we also have surprising results. In fact, differently from the LeNet results, we observe that with the ResNet-20 the MIME algorithms fail at improving FEDAVG. This is particularly evident on CIFAR-100, where there is actually a big performance drop both with Mime and MimeLiteMom. Even MimeMom only achieves results comparable to FEDAVG. Although we may conjecture that our grid search may be not broad enough if that were the case it would be a behaviour indicative of high sensitivity to hyper-parameters choice (we also confirmed our results by using the original Jax code provided by the authors). Conversely, FEDHBM effectively improves FEDAVG by a large margin also on ResNet-20, both on CIFAR-10/100.

## B.4 ADDITIONAL ABLATIONS

**Using Nesterov Accelerated Gradient** Table 11 includes results with the Nesterov Accelerated Gradient variant of momentum. As it is possible to notice, in our setting this variation does not lead to significant differences overall. These results are in line with those obtained by (Ozfatura et al., 2021).

**Ablation on momentum term** In figure 6 we show an ablation study on the $\beta$ term in our momentum factor $\hat{\beta}_i = {}^{\beta C}/_{J_i}$. In standard momentum, the $\beta$ factor indicates the decaying rate of past gradients, and for this reason, it is usually a number lower than 1, with common values being $\{0.9, 0.99\}$. Intuitively, a value closer to 1 will make the momentum accumulate past directions, while $\beta = 0$ means no momentum at all. As graphs show, setting $\beta = 1$ leads to more stable

Table 11: Test accuracy (%) comparison of several SOTA FL algorithms on our **cross-silo** setting. Best result is in **bold**, second best is underlined.

| METHOD | CIFAR-10 | | CIFAR-100 | | SHAKESPEARE | |
|---|---|---|---|---|---|---|
| | NON-IID | IID | NON-IID | IID | NON-IID | IID |
| FEDAVG | $66.12 \pm 0.32$ | $83.11 \pm 0.34$ | $35.56 \pm 0.24$ | $49.74 \pm 0.22$ | $47.31 \pm 0.10$ | $47.08 \pm 0.17$ |
| FEDPROX | $66.12 \pm 0.32$ | $83.11 \pm 0.34$ | $35.48 \pm 0.30$ | $49.86 \pm 0.22$ | $47.30 \pm 0.10$ | $47.07 \pm 0.17$ |
| SCAFFOLD | $74.83 \pm 0.20$ | $82.93 \pm 0.25$ | $45.50 \pm 0.12$ | $49.41 \pm 0.40$ | $50.25 \pm 0.10$ | $50.13 \pm 0.10$ |
| FEDDYN | $70.93 \pm 0.18$ | $83.52 \pm 0.12$ | NaN | $\underline{51.95} \pm 0.17$ | $50.72 \pm 0.12$ | $50.80 \pm 0.16$ |
| ADABEST | $66.12 \pm 0.36$ | $83.11 \pm 0.38$ | $35.56 \pm 0.26$ | $49.74 \pm 0.25$ | $47.31 \pm 0.10$ | $47.08 \pm 0.17$ |
| MIME | $75.08 \pm 0.55$ | $83.13 \pm 0.46$ | $36.31 \pm 0.49$ | $50.87 \pm 0.36$ | $48.29 \pm 0.16$ | $48.49 \pm 0.15$ |
| FEDAVGM | $67.58 \pm 0.27$ | $83.60 \pm 0.31$ | $35.22 \pm 0.33$ | $50.68 \pm 0.25$ | $50.00 \pm 0.03$ | $50.41 \pm 0.08$ |
| MIMEMOM | $80.95 \pm 0.40$ | $83.11 \pm 0.20$ | $48.17 \pm 0.68$ | $50.60 \pm 0.11$ | $48.46 \pm 0.19$ | $48.89 \pm 0.25$ |
| MIMELITEMOM | $78.79 \pm 0.38$ | $83.23 \pm 0.29$ | $46.00 \pm 0.30$ | $50.66 \pm 0.10$ | $49.10 \pm 0.38$ | $49.39 \pm 0.32$ |
| **FEDHBM (ours)** | $\underline{81.71} \pm 0.15$ | $\mathbf{83.83} \pm 0.14$ | $\mathbf{50.41} \pm 0.51$ | $\mathbf{51.99} \pm 0.45$ | $\mathbf{51.33} \pm 0.08$ | $\underline{51.36} \pm 0.19$ |
| **FEDHBM - Nesterov (ours)** | $\mathbf{81.93} \pm 0.15$ | $\underline{83.77} \pm 0.13$ | $\underline{50.25} \pm 0.60$ | $51.40 \pm 0.64$ | $\underline{51.30} \pm 0.10$ | $\mathbf{51.38} \pm 0.20$ |

convergence. Note that this is not in opposition to the common practice of having a momentum factor $\beta \in [0, 1)$, since our momentum term decays with $\hat{\beta}_i$ rate, so as long $^C/_{J_i} < 1$ it is safe to set $\beta = 1$.

**Measuring the effect of heterogeneity** Figure 7 presents an analysis of the effect of heterogeneity on i) final model quality (left) and ii) convergence speed (right). The experimental results, while confirming that it is crucial to perform some form of drift control during local optimization, show that momentum methods handle extreme heterogeneity scenarios better than methods that rely on stochastic variance reduction, as SCAFFOLD . Let us notice that the considered algorithms are robust w.r.t non-extreme heterogeneity: this underlines the need for algorithms that do not sacrifice communication efficiency for robustness to non-iidness. The right part of the figure shows that heterogeneity has a strong effect also on convergence speed. In line with the results on the left graph, MIMEMOM and FEDHBM are the fastest when facing the pathological case of $\alpha = 0$. Surprisingly, MIMEMOM is not significantly faster than FEDAVG and FEDAVGM in non-extremely heterogeneous scenarios; indeed it is slower if taking into account the communication overhead. In all cases FEDHBM performs best, demonstrating high robustness to heterogeneity from both the considered perspectives.

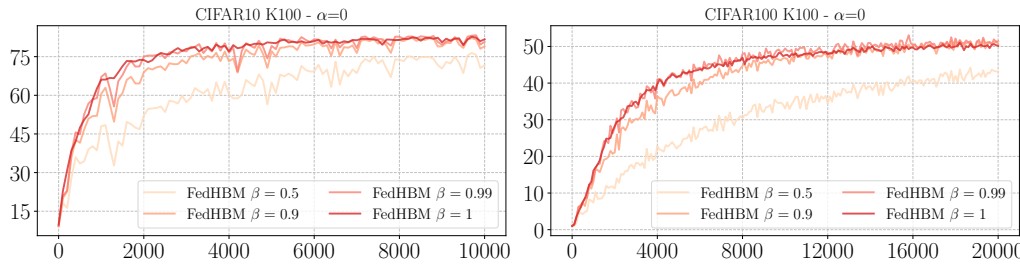

Figure 6: Ablation study on $\beta$ values for FEDHBM on CIFAR-10/100 in our cross-silo setting

**Additional experiments with learning rate decay** This paragraph has been added in response to a reviewer's concern about the use of learning rate schedules in our experimental validation. In this paper, we claimed that using learning rate schedules can cause unfair comparison, as different algorithms exhibit very different convergence rates, especially in non-iid settings. Let us also point out that many well-established works in FL do not use learning rate schedules (McMahan et al., 2017; Li et al., 2020; Hsu et al., 2019; Karimireddy et al., 2020; 2021), while some others do (Acar et al., 2021). Figure 8 shows the accuracy curves of the best FL algorithms from table 10, using a learning rate decay with decay coefficient fine-tuned for each algorithm, searched in the range $\{0.999, 0.9992, 0.9995, 0.9999\}$. For all the algorithms, the best learning rate decay turned out to be 0.9999. Comparing with performances without learning rate decay reported in table 10, it is possible to notice that: i) the use of learning rate, in general, does not change the relative performance of the algorithms and ii) the use of learning rate decay, in these settings, does not help convergence,

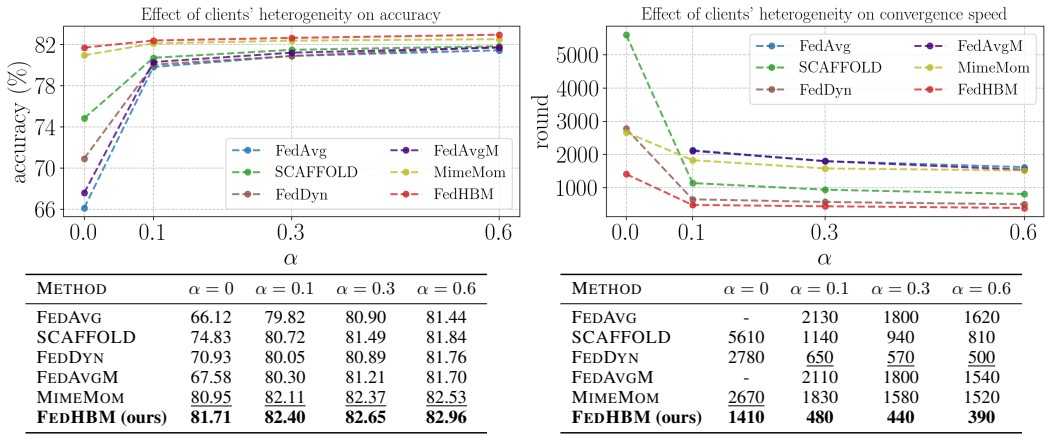

Figure 7: Ablation study on the effect of several degrees of heterogeneity on performance of SOTA algorithms and FEDHBM on CIFAR-10. The left figure shows the final accuracy reached by algorithms, while the right figure shows the number of rounds needed to reach 70% of absolute accuracy. The tables show the values depicted in the respective picture above. The best results are in **bold**, second best are in underlined.

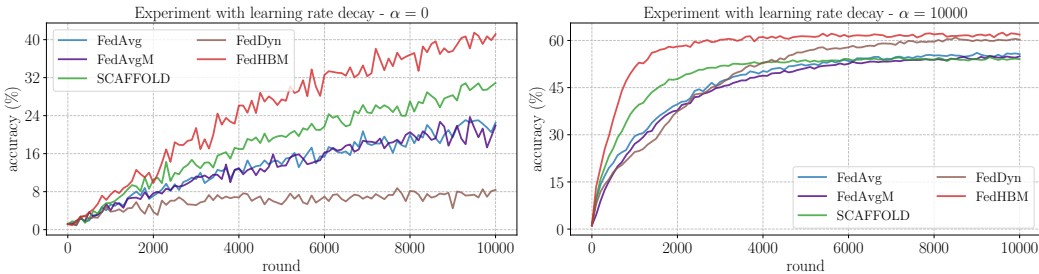

Figure 8: Experiments with learning rate decay of SOTA algorithms and FEDHBM on CIFAR-100. The decay coefficient has been searched in the range $\{0.999, 0.9992, 0.9995, 0.9999\}$ for each algorithm.

especially in non-iid scenario, where the performances are degraded w.r.t. not applying any schedule. This is motivated by the fact that a large number of rounds is needed to achieve convergence, and probably the simple decay strategy, which we adopt from Acar et al. (2021), is not optimal to practically give an advantage. Other learning rate schedules may be more appropriate, but this largely expands the needed hyperparameter search, considering that it must be searched separately for each algorithm.

