# OpenReview forum: "Communication-Efficient Heterogeneous Federated Learning with Generalized Heavy-Ball Momentum"
_ICLR.cc/2024/Conference — Submitted to ICLR 2024_

### Official Review · Reviewer_8a6j · 2023-10-16

**Soundness:** 2 fair
**Presentation:** 3 good
**Contribution:** 1 poor
**Rating:** 3
**Confidence:** 5

**Summary:**

This paper proposes to adopt the generalization of heavy-ball momentum in FL which uses an uncertain interval according to the storage state of each local client to construct the momentum term (the current state minus the last local state). No theoretical analysis of convergence or generalization is provided. Experiments on CIFAR-10/100, Shakespeare are conducted to validate its efficiency without learning rate decay.

**Strengths:**

1. The proposed shift window can help to reduce the communication requirements.
2. This paper proposes a summary of the classical momentum-based method in FL, i.e., FedCM, FedADC, and MimeMom.
3. The experiments are widely conducted on several setups to validate the efficiency.

**Weaknesses:**

1. In Eq.(5), what is the MVR term? Could the author explain this in detail? Its first part is a sum of $J_i$-step updates, while its second part is a sum of $j$-step updates. I know these two parts come from the additional term from $\theta_{i,j}^t-\theta_i^{t-\tau_i}$ to $\theta^{t-1}-\theta^{t-\tau_i-1}$, but how does it perform as a variance reduction? The $u$ term is the update performed in each local iteration, so the MVR term is the difference between $j$ local updates and $J_i$ local updates where $j\neq J_i$. Please provide the equivalent form in [1] and demonstrate its variance reduction efficiency in the main text.

   [1]: Momentum-based variance reduction in non-convex sgd

2. The authors claim that they propose the GHB form in Table 1. However, the same updates have been studied in [2] which adopts a multi-step momentum. By setting the specific coefficients as some fixed constants, [2] performs the same updates as the global GHB. Actually, the claimed global GHB in this paper is only a special case of [2]. It also provides some convergence analysis to understand more complicated cases. Therefore, the contribution of this paper seems to only extend the global GHB to the local GHB without any analysis of optimization or generalization, which greatly reduces the novelty of this paper.

    [2]: Enhance local consistency in federated learning: A multi-step inertial momentum approach

3. In the experiments, it indicates that the non-iid dataset split adopts a $\alpha\rightarrow 0$. While in the appendix, it shows $\alpha=0$. Please unify this statement.

4. The results are incomplete. For instance, in Table.2, 3, 4, there are no results of FedCM and FedADC. However, in Figure.3, the curves of FedCM are stated as an ablation study. As the very important baselines of this paper, the comparison with FedCM and FedADC is very necessary, which are also two SOTA methods among the momentum-based methods. Authors should add their performance tests to this paper.

5. The experiments do not adopt the learning rate decay. On page 15 in the appendix, the authors claim that all experiments do not use a learning rate scheduler for simplicity. However, this will significantly reduce the performance of some algorithms and make comparisons unfair. For instance, the ADMM-based method, i.e. FedDyn, requires to be optimized well enough on the local client. Otherwise, dual variables will be updated with very large biases. A similar phenomenon also happens in SCAFFOLD and even FedCM. I think this is the main weakness of the experiment in this paper. All results only reflect the phenomenon under a fixed learning rate. While in the current machine learning, the learning rate decay in non-convex optimization is very important and one of the major concerns. Although changing the learning rate complicates comparisons, this kind of comprehensive study can broadly reflect the performance of the proposed method.

**Questions:**

Thanks for this submission with the FedGBH method. My main concerns are stated in the weaknesses, mainly including the novelty with repetitive parts from the previous studies, and the lack of baselines and hyperparameter settings.

---

> ### Author Response · Authors · 2023-11-16
> **Response to W1, W3, W4, W5 (1/2)**
>
> We thank the reviewer for the feedback. In the following, we address the reviewer’s remarks on the weaknesses (W).
> * **W1:** In our initial analysis of the second term in Eq. (5), we drew parallels with the concept presented in [1], perceiving it as a variance reduction mechanism due to observed similarities. We appreciate the Reviewer's keen observation regarding the distinction between these two terms. This insight has led us to a more accurate interpretation and, consequently, a revision in our manuscript. We now clarify that this term in our formulation is intended to penalize the direction of the last updates at round $t - \tau_i$ ​, in contrast to the progressive update of local steps at the current round $t$.
> Although at the moment we cannot formally prove that this term acts as variance reduction, we investigated its importance in the ablation study section, specifically in Figure 2, where it is evident that it provides additional speedup w.r.t Local-GHB and reaches better accuracy (see fig. 3 and table 10).
> * **W3:** Thanks for spotting the typo, we fixed it.
> * **W4:** In the first part of section 4.3 we discuss how both FedCM and FedADC are equivalent formulations of GHB with fixed $\tau=1$ (and hence, extra communication): this is shown analytically in table 1, but also experimentally in figure 3, where the performance of GHB $\tau=1$ and FedCM are shown to be equivalent. This is the reason why we didn’t include them in the main results, as we specified in the first paragraph of section 4.1. However, since this may not be immediately obvious to the reader, we added the values of FedCM and FedADC in the main tables, as you suggested, to dispel any doubts about the completeness of our experimental validation.
> * **W5:** The reason why we did not adopt a learning rate decay is because we considered that this would have caused unfair comparison since algorithms have very different convergence speeds. Please notice that this way of conducting the experiments is in line with most existing well-established works in FL [3,4,5,6,7].  However, to demonstrate that our findings hold even when adopting learning rate decay, we added in the supplementary an additional section showing the accuracy curves for the iid and non-iid training of CIFAR-100 with ResNet-20, with full hyperparameter search the learning rate decay and algorithm-specific hyperparameters. For ease of comparison, we also added in Table 10 the results for iid training of CIFAR-10/100 without learning rate decay.

---

> > ### Author Response · Authors · 2023-11-16
> > **Response to W2 (2/2)**
> >
> > * **W2:** Thanks for suggesting this work. We were not aware of it and we will discuss it in the paper. We tried to cover the most relevant works in NeurIPS/ICLR/ICML venues. However, [2] seems not to be a peer-reviewed work, as we only found it on arXiv. Please kindly provide us with the correct reference if we are mistaken so that we can appropriately cite it. Please notice that [2] does not perform the same updates of our FedHBM, as we motivate below.  \
> > **Important differences with [2]:** \
> > Their $\delta_{t}=-(x_{t} - x_{t-1})/K$ (line 6 of their algorithm 1) takes the difference of models at two subsequent rounds. To perform that operation, the server must send both models to the client, which sacrifices communication efficiency unless all clients participate in each round, which is not a realistic FL scenario. Their clients also need to keep track of this multiple $\delta_{t-j}$, with linear increase (w.r.t. to model parameters) of the needed client storage. This also exponentially increases the number of hyperparameters $\alpha_{j}, \beta_{j}$ to be tuned. In fact, for practicality, authors fix the number of deltas to two, and so four hyper-params to be tuned. For these reasons, we believe that actually [2] is a special case of GHB with hand-tuned $\tau$ (in practice set to two in [2]). The term “multi-step” just refers to the adoption of the Nesterov momentum formulation instead of the common one adopted by us. \
> > Conversely, the momentum term in FedHBM is computed over a window, whose size is automatically dictated by the client's participation. This requires keeping only one model stored on the client side, and a single momentum factor $\beta$ to be tuned. Our ablation 4.3 extensively tests the effect of the window size $\tau$, highlighting the importance of $tau>1$ and the optimality of the self-tuned $\tau_i$ for communication efficiency. Crucially this is an important difference that sets us apart from [2]. \
> > Moreover, we show that the FedHBM update rule implicitly contains a momentum variance reduction term. As already mentioned, it gives an additional speedup, especially in the most heterogeneous scenarios. \
> > **Concerning the experimental validation.** \
> >     *  **Small scale vs comprehensive evaluation.** The work [2] only validates small-scale computer vision tasks, with only one architecture, which is not specified. Conversely, in our work, we considered both computer vision and NLP tasks, multiple architectures (LeNet, ResNet-20, MobileNetv2), and extended also to large-scale real-world datasets. Such extensive evaluations are important contributions since they show issues not noticed by previous works.
> >     * **High vs extremely high heterogeneity.** In [2] authors only test under non-extreme heterogeneity (e.g. as low as $\alpha=0.1$), while we showed that in those non-extreme scenarios, even simple baselines (as FedAvgM) are satisfactory (see fig. 1 in our paper).
> > As such, those scenarios do not render the robustness issues of existing algorithms. Conversely, in our work, we extensively tested the robustness of the algorithms in the most difficult scenarios ($\alpha=0$) which are generally not covered by other works. Such a study on the robustness against extreme heterogeneity is indeed one of the contributions of our work.
> >     * **Cross-silo and cross-device.** [2] does not take into account cross-silo and cross-device scenarios, and how their algorithm performs in both cases. We took special care about exploring how our solution works under such conditions, as we presented extended empirical evidence to prove that FedHBM works also in a scenario it was not originally designed for. Large-scale experiments in Table 4 prove that a participation ratio as low as 0.5% is enough to see the practical advantages of our method over the state-of-the-art. As these analyses are important to evaluate the versatility of an FL algorithm for real-world scenarios, they should be considered as a strength of our work.
> >     * **Reproducibility.** Lastly, [2] lacks necessary implementation details, which together with the absence of official code makes results not easily reproducible. Conversely, we devoted much space in the supplementary to provide the reader with a comprehensive description of the experimental setup and hyperparameter search for all the algorithms we tested (see section B.1, especially table 7). We also attach our code, to underline our commitment to full reproducibility.
> >
> > For these reasons, the novelty of our work is by no means diminished by [2], nor the contribution we bring to the FL research community, as we provided value in focusing on underexplored cases, showing failure cases of existing SOTAs, unifying existing momentum-based FL works in one framework (GHB), and finally proposing and extensively validating our new FedHBM algorithm.
> >
> > We hope that our answer addresses your concerns and that you consider revising your rating of our work for an “accept” decision.

---

> > > ### Author Response · Authors · 2023-11-16
> > > **References**
> > >
> > > We are available for answers to any remaining questions on our work.
> > >
> > > [1] Momentum-Based Variance Reduction in Non-Convex SGD, NeurIPS 2019\
> > > [2] Enhance Local Consistency in Federated Learning: A Multi-Step Inertial Momentum Approach, arXiv 2023\
> > > [3] Communication-efficient learning of deep networks from decentralized data, PMLR 2017\
> > > [4] Measuring the effects of non-identical data distribution for federated visual classification, arXiv 2019\
> > > [5] Federated optimization in heterogeneous networks, PMLR 2020\
> > > [6] Scaffold: Stochastic controlled averaging for federated learning, PMLR 2020\
> > > [7] Breaking the centralized barrier for cross-device federated learning, NeurIPS 2021

---

> > > > ### Comment · Reviewer_8a6j · 2023-11-22
> > > > **Response to the rebuttal**
> > > >
> > > > I have read all rebuttals from the authors and thanks for the explanation. I have to strengthen some unsolved concerns:
> > > >
> > > > 1. Variance reduction is a rigorous algorithm in the optimization. The reason "perceiving it as a variance reduction mechanism due to observed similarities" is insufficient. I'm not sure if this calculation is equal to the VR algorithm mentioned in the article, this needs further confirmation in detail. It is best for the authors to provide some proof to show that this term has variance-reducing properties or asymptotic properties.
> > > >
> > > > 2. The learning rate is constant. The authors claim that classical works adopt the constant learning rate. However, the works mentioned were mostly published 2021 years ago. Actually, as the baseline in this paper, FedCM and FedADC all adopt the learning rate decaying in their experiments. As I said in the comments, the experimental phenomena of constant learning rate and learning rate with decay term are very different. Therefore, I believe that the comparison in this article is biased to the real results. On the basis of the constant learning rate experiment, I think that the experiment of learning rate decay should at least be tested like the baseline papers.
> > > >
> > > > Thanks for the good submission and I will keep the current score for the above main concerns.

---

> ### Author Response · Authors · 2023-11-23
> **About the remaining concerns**
>
> Thanks for the feedback. About the remaining concerns:
> 1. We appreciated your observation, and the rebuttal version provides a different intuition about that term. The current version does not claim that the term is equal to the algorithm previously mentioned or a variance reduction term.
> 2. **About constant learning rate:** as the reviewer suggested in its initial review, we added these extra experiments in our revision (in both iid and non-iid scenarios, in appendix B.4, figure 8). As suggested and following the practice mentioned in FedDyn and FedCM, we used an exponential learning rate decay by decreasing the local learning rate by a $0.9999$ factor at each round. This value has been searched to be the optimal one among $\\{0.999, 0.9992, 0.9995, 0.9999\\}$. \
> Our results are clear in proving that adopting a learning rate decay does not change the comparisons between algorithms in the setting tested ($\alpha=0$ and $\alpha=10000$) and showing that our method outperforms all the baselines.\
> We remark that all the works considered in our evaluation represent the state-of-the-art methods generally used as baselines in all the Federated Learning papers.
>
> Given the requested experiments, it is unclear to us what is missing to fully convince the reviewer that our results are valid and our method has been fairly and extensively compared against the baselines.

---

### Official Review · Reviewer_Yr1w · 2023-11-04

**Soundness:** 2 fair
**Presentation:** 2 fair
**Contribution:** 1 poor
**Rating:** 3
**Confidence:** 3

**Summary:**

The paper adjusts and combines two existing momentum-based methods applied to federated learning. The first method (acceleration) is the heavy ball method (i.e., Polyak's momentum). The second method is a variance reduction technique [1]. The adjustment for these two methods is to parameterize the time step for the previous iterate of the momentum term for each client. For example, with the heavy ball, the momentum term becomes: $\frac{\beta}{\tau_i}(\theta^{t-1} - \theta^{t-{\tau_i}-1})$ where $\tau_i$ is the time step parameter for client $i$. Experiments are conducted on both iid and non-idd federated learning datasets.

[1] Cutkosky, A. and Orabona, F. Momentum-based variance reduction in non-convex SGD. 2019.



Update: I thank the authors for replying to my review/questions. I have also read through the other reviews and responses. I will maintain my score.

**Strengths:**

1. Explores the use of momentum for federated learning.
2. The paper is easy to read. However, the writing needs to be improved.

**Weaknesses:**

1. The novelty of this paper is limited. As mentioned in the summary, two existing momentum-based techniques are additively combined with the modification to adjust previous iterate time step parameter. In my opinion, to call the proposed formulation generalized heavy-ball momentum is somewhat a far-stretch.
2. Some theoretical analysis not provided. Although averaging has been applied, it is not clear why having this gap or "window" is desirable in general and particularly in the federated learning setting. Motivation/intuition needs to be given.
3. Not clear if the results are reproducible since code is not given.
4. Missing some related work:
   * Xin, R. and Khan, U. Distributed heavy-ball: a generalization and acceleration of first-order methods with grading tracking. 2018.
   * Das, R. et al. Faster non-convex federated learning via global and local momentum. 2022.
   * Kim, G. et al. Communication-Efficient Federated Learning with Acceleration of Global Momentum. 2022.

Additional: labels missing on graphs.

**Questions:**

Very little information is provided on setting $\tau_i$. Section A of supplemental states: $\tau_i \rightarrow \tau = 1/C$ where $C$ is the number of clients. If this is the case, then the term appears negligible under large scale setting.

---

> ### Author Response · Authors · 2023-11-16
>
> We thank the reviewer for the feedback. In the following, we address the reviewer’s remarks on the weaknesses (W) and questions (Q).
> * **W1:** First, let us restate the novelty of our approach. The generalization we claim is based on the fact that, while standard momentum and its declinations in FL use the updates that occurred in the last round, we consider a larger window (GHB) across multiple rounds. While this modification results in a simple implementation, we showed that enlarging this window to a length $\tau>1$ is crucial to developing the robustness against extreme heterogeneity (see fig. 3). On top of this idea, we developed communication-efficient variants of GHB, namely Local-GHB and our proposed FedHBM, which self-control the window length according to client participation and this is optimal both in terms of window length and communication costs (see again Fig 3).
> As an additional point, we frame existing momentum FL algorithms inside this generalization, and show how they are special cases (see Table 1): in this sense, our work brings some order to the literature on momentum in FL, which is part of our contribution.
> * **W2:** We acknowledge that a formal proof could strengthen our paper. However, we also argue that theoretical convergence rates may only render an incomplete picture, whereas extensive experiments may reveal more than an elaborate yet very specific proof. As a matter of fact, in our experiments, empirical evidence demonstrated that in extreme cases state-of-art algorithms, despite being supported by formal guarantees, happen to actually fail (e.g. see Table 4). \
> **Motivation behind a window $\tau > 1$.** \
> The main intuition behind our method is that the trajectory of the server updates over a window $\tau>1$ provides a better estimate for the momentum term in a federated setting. Intuitively, as $\tau$ increases,  the momentum term increasingly incorporates information from a broader range of clients. A key observation is that when $\tau$ equals the average period length (e.g. $\tau = \frac{1}{C}$), under uniform client sampling, the momentum term contains the information on the global distribution and hence is optimal.  We experimentally verified this hypothesis, demonstrating its validity in practice as we showed by purposely varying $\tau$ in Figure 3. We revised the manuscript to better describe these details. \
> **Is it always beneficial?** \
> Yes. According to our experiments, **in FL this is always beneficial** since it improves even in the iid scenario, where the simple FedAvgM is often a strong baseline. However, its effect is most noticeable in extreme non-iid scenarios where computing the momentum over a larger window mitigates the effects of heterogeneity and partial participation (see Fig. 2 and 3)..
>
>
> * **W3:** We take reproducibility into great consideration since we provided a precise algorithmic description in Algorithm 1 and accurately reported the datasets, the split procedures, the hyperparameters, and the framework used (see section B and in particular table 7). We did not attach the code to the original submission, but following your suggestion we are attaching it now, we hope this may dispel your concerns.
> * **W4:** Thanks for reporting to us these works we were not aware of, we will take the time to go into details and we will include them in our revision of the related works.
> * **Q1:** The formula reported is correct, but the term C is not the total number of clients. Rather, it is the participation ratio, i.e., the portion of clients that participate in each round (i.e., $0<C\leq1$). In practice, $\frac{1}{\tau_i}$ term has the meaning of an averaging factor, and so it fits the number of contributions it averages. This is perhaps clearer by looking at the GHB row in Table 1 (right), which contains the expansion for a fixed $\tau$.
> Let us remark that in Local-GHB and FedHBM, $\tau_i$ is not hand-tuned, but it is instead determined stochastically by client participation: in practice, under uniform sampling, on average each client automatically considers a window of length $\tau_i \approx \frac{1}{C}$.
> We showed the optimality of  $\tau_i$ under uniform client sampling in figure 2, which also crucially allows us to be communication-efficient, whereas hand-tuning $\tau$ requires an overhead of $1.5\times$ w.r.t. FedAvg.
>
> We hope that our answer addresses your concerns and that you consider revising your rating of our work for an “accept” decision.

---

### Official Review · Reviewer_ZLT2 · 2023-11-04

**Soundness:** 3 good
**Presentation:** 3 good
**Contribution:** 2 fair
**Rating:** 6
**Confidence:** 4

**Summary:**

This paper studies the problem of system and statistical (data) heterogeneity in the context of Federated Learning. Specifically, the authors propose a novel federated algorithm that utilizes a generalization of the the heavy-ball momentum method on the client side to achieve improved final accuracy and convergence speed in non-iid regimes both in cross-silo and cross-device settings. Extensive numerical results are presented both on academic data set as well as on real-world applications. These experiments showcase the superiority of the proposed FedHBM (and practical variation of this algorithm) in terms of communication cost, accuracy, and convergence speed compared to state of the art federated methods.

**Strengths:**

-The paper studies an important problem in the area of Federated Learning namely the combination of system and statistical heterogeneity.

-The paper is well structured and easy to follow.

-A simple and intuitive algorithm is proposed that relies on a generalization of the heavy-ball momentum on the client side.

-Both cross-silo and cross-device settings have been explored in the experiments. Both academic and real-world datasets have been studied.

-Extensive ablation study exhibits the effects of $\tau$ (captures which model is used for calculating the momentum i.e. from how many rounds in the past this model is chosen) on the performance of the algorithm.  Additionally, a practical variation of the algorithm has been presented without additional communication requirements with strong performance.

**Weaknesses:**

-The main weakness of the paper lies on the absence of theoretical results.

-The proposed algorithm is a rather simple generalization of the heavy-ball momentum and as a result the novelty is limited.

-In the cross-device setting the algorithm requires a 'proper' initial model to achieve the required improvement.

-In table 2 it seems that FEDDYN achieves higher target accuracy (90%) in the seemingly more challenging cross-device setting. I would appreciate it if the authors could elaborate on that.

-In the plots of Figure 3 and Figure 4 (a) it is hard to distinguish between some of the methods. I would recommend to either utilize different colors or increase the scale.


Minor issues

- Page 3, paragraph 3, in"..existing algorithms can be express as special.." replace express by expressed.
- Page 6, paragraph 4, in "$\alpha = 10k$" define $k$.
- In table 5  for $C\approx 0.5$ FEDAVG's performance is in bold which appears to be a typo.

**Questions:**

See weakness section.

---

> ### Author Response · Authors · 2023-11-16
>
> We thank the reviewer for the feedback, we are pleased to notice that the reviewer appreciated our work. In the following, we address the reviewer’s remarks on the weaknesses (W).
> * **W1:** We acknowledge that a formal proof could strengthen our paper. Certainly, a theoretical proof of convergence is one way to support the claims of a method, but it is not the only way. Here we chose to have an empirical approach, where we test extensively FedHBM and existing methods in a wide range of tasks, extreme scenarios, and real-world cases, as we are pleased you appreciated. By doing so we provide evidence that our method consistently outperforms existing methods which are supported by articulated formal proofs [1,2]. We also argue that theoretical convergence rates may only render an incomplete picture, whereas extensive experiments may reveal more than an elaborate yet very specific proof. As a matter of fact, in our experiments, empirical evidence demonstrated that in extreme cases state-of-art algorithms, despite being supported by formal guarantees, happen to actually fail (e.g. the case of FedDyn diverging already known in the literature, and SCAFFOLD and MimeMom in our table 4).
> * **W2:**  We agree that our algorithm is actually simple to implement, and we consider it as a strength. In fact, more often than not, very complex and articulated methods require careful tuning, which may be case-specific, and are prone to introducing coding errors. On the contrary, we think that _simplicity is the ultimate sophistication_ [Leonardo Da Vinci].
> We also believe that simplicity should not be considered as an indication of a lack of novelty.
> To the best of our knowledge, we are the first to propose a generalization of the heavy-ball momentum computed over a period $\tau>1$, and we showed that it has a significant impact on performance in FL (figure 3). Within this framework (GHB), FedHBM allows to capture **locally** (i.e., on the client side) global information that encompasses more than one round, without communication overhead.
> We analyzed its practical advances under a wide range of scenarios, to support our claims of FedHBM being superior to the state-of-the-art FL methods. These results, as well as the practical demonstration of the failure of well-established FL algorithms in extreme non-iid cases, are themselves a novelty that adds to the knowledge base of the community, and points to new problems to be considered.
> * **W3:** Results in Table 4 use the formulation proposed in algorithm 1, and nonetheless our method is in most cases the best, so the practical variation is not strictly necessary to achieve good performance. Our point is that, although our algorithm is primarily designed for cross-silo scenarios, it can still be used for cross-device scenarios, as the variation we proposed can help obtain a speedup earlier (see Figure 4).
>
>
> * **W4:** Table 2 reports the number of rounds to reach a target accuracy w.r.t the accuracy of centralized training (reported in table 6), hence the lower the number the faster the algorithm. In a cross-device scenario, FedDyn reaches 90% of the accuracy reached by centralized training in 7600 rounds, which is worse than our FedHBM, which needs only 6510 rounds to reach the same result.
> Namely, table 2 reports the speed while Tables 3 and 4 report the final performance of the algorithms. Together, these results show that our algorithm is the most communication efficient and that it yields the best model quality.
> * **W5:** Thank you for your suggestion, we updated the manuscript accordingly.
> * **Minor issues:** Thanks for spotting the typos, we fixed them. On page 6, par. 4, $k$ simply stands for $\times1000$. As we understand it can generate confusion, we replaced it with $\alpha=10.000$
> We hope that our answer addresses your concerns and that our revision will contribute positively to the evaluation of our work.
>
>
> [1] Scaffold: Stochastic controlled averaging for federated learning, PMLR 2020 \
> [2] Breaking the centralized barrier for cross-device federated learning, NeurIPS 2021

---

> > ### Comment · Reviewer_ZLT2 · 2023-11-23
> > **Post Rebuttal**
> >
> > I thank the authors for addressing my concerns and for implementing minor revisions to their draft. After carefully reading the authors rebuttal and the comments from the other reviewers I consider this paper to lie on the acceptance threshold and I slightly lean towards acceptance. Although the experimental results appear promising, the main drawbacks of the paper (absence of theoretical results and limited novelty especially in light of "Enhance Local Consistency in Federated Learning: A Multi-Step Inertial Momentum Approach") remain.

---

> > > ### Author Response · Authors · 2023-11-23
> > > **Response to Post Rebuttal**
> > >
> > > We are glad the reviewer put care into reading the rebuttal and our revisions.
> > >
> > > We acknowledge the value of formal proofs, but we showed that often they render an incomplete picture, as even cornerstone FL approaches clearly fail in some scenarios: most notably, this is shown in table 4 for real-world experiments. We believe that the absence of a formal proof in an experimental paper should not be penalized more than the lack of extensive experiments in theoretical papers (like usually it is).
> > >
> > > About the novelty with respect to [3], we extensively discussed the work mentioned [here](https://openreview.net/forum?id=oJ1tx3fXDA&noteId=DVdkN8tn1N) and highlighted many of the differences that crucially set us apart from them (most notably, their method is not communication efficient, which is the very focus of our work). Moreover, we stress that the work mentioned has not been peer-reviewed. We have shown that the novelty of our work is by no means diminished by [3], nor the contribution we bring to the FL research community
> > >
> > > [3] Enhance Local Consistency in Federated Learning: A Multi-Step Inertial Momentum Approach, arXiv 2023

---

### Official Review · Reviewer_wwM7 · 2023-11-06

**Soundness:** 3 good
**Presentation:** 3 good
**Contribution:** 3 good
**Rating:** 8
**Confidence:** 4

**Summary:**

Th paper proposes a new Federated Learning (FL) algorithm based on heavy-ball momentum, designed to be more robust to statistical data heterogeneity without significant communication cost, compared to state-of-the-art FL techniques. The method proceeds by computing local momentum at the client level, and results in a novel algorithm called FedHBM (heavy ball momentum) which generalizes existing momentum based FL algorithms. The empirical properties of the method are illustrated on vision and NLP tasks, and compared favorably to competitors.

I have read the authors comments which clarified some important aspects of the experiments performed. Accordingly, I upgraded the "soundness" rate to 3-good, as well as my overall rating to 8.

**Strengths:**

- The paper is very well written, clear and dynamic. Both mathematical statements and intuitive explanation are very clear and easy to follow.
- The proposed methodology, which efficiently exploits participation of clients at multiple optimization rounds to estimate local momentum without increasing communication cost is novel and clever.
- The relation with prior work is very clear and thoroughly discussed, in terms of general properties, precise mathematical formulation and empirical performance
- The limitations of the FedHBM are discussed, particularly in the case of high cross-device settings, and alternatives are considered to alleviate them.

**Weaknesses:**

- The paper does not provide any insight on how to set the values of the step sizes \eta and \beta. In addition, the authors do not mention how they set it for their experiments. This is a major limitation to practical use of the algorithm.
- In relation with aforementioned issue, the authors do not discuss how other hyperparameters were set for competitors, thus comparisons are not easy to interpret (was their hyperparameter optimization for all algoritms ?)
- Overall, the rationale for the choice of the results presented (nb of rounds to reach accuracy level, final model accuracy) is not completely clear. For instance, the computational cost per round is not discussed, so it is difficult to understand the implication of these results. In addition, the non-iid setting is not clear. The authors mention using a Dirichlet distribution, which suggests drawing vectors of probabilities to assign samples to clients, but I don't see how this would lead to heterogeneity (only size imbalance between clients).

**Questions:**

- How were step sizes \eta and \beta set in the experiments ? How about similar hyperparameters of competitors ?
- Is the computational cost per round similar between algorithms ?
- What does final mean ? Are all algorithms stopped after 10000 rounds ?
- Could you provide more insight on how you designed the non iid setting ? How do you split the data ? How do you use the Dirichlet distribution ? Is it used to draw probabilities of assignment to each client ? Then, how do you assign samples to clients ? It should depend on the sample value, and I don't see it in your explanations.

---

> ### Author Response · Authors · 2023-11-16
>
> We thank the reviewer for the feedback, we are pleased to notice that the reviewer appreciated our work. In the following, we address the reviewer’s remarks on the weaknesses (W) and questions (Q).
> * **W1-W2-Q1:** For clarity, let us restate the hyperparameters involved in our method. $\eta$ is the server learning rate, $\eta_l$ is the client learning rate, $\hat{\beta_i}$ is our momentum factor, defined as $\frac{\beta C}{J_i}$, where $J_i$ is the number of local steps and $C$ is the participation ratio ($0<C\leq1$). The hyperparameters to be tuned are $\eta$, $\eta_l$, and $\beta$.
> For each method, the hyperparameters have been searched via a grid search. As we take into great consideration reproducibility, in Table 7 we reported our extensive hyperparameter search for all the algorithms, indicating in bold the best values.
> * **W3-Q2:** We assumed the computational cost per round to be equal for all algorithms, as usually, communication cost is the bottleneck. By doing so, we acted conservatively, as Mime, MimeMom, and MimeLiteMom actually require multiple forward and backward passes on the local dataset, and so require much more computations (we refer to [1] for additional details on their algorithm). All the other algorithms (including ours) have similar computational costs since they just require addition/subtraction to model parameters, which are not expensive. As a matter of fact, this is reflected in the wall-clock training time of our simulations, as indicated in section B.2, under the paragraph “Implementation details and practicality of experiments”.
> * **W3-Q4:** Regarding the non-iid setting, we followed the practice in [2], which is common in FL works. In practice, it consists of drawing class probabilities for each client using a Dirichlet distribution with parameter $\alpha$. The lower the value of $\alpha$, the higher the class imbalance we introduce into local datasets. We acknowledge your suggestion, and we added some more words in the supplementary B.1 to better explain the procedure in [2].
>
>
> * **Q3:** _final model quality_ means the top-1 accuracy over the last 100 rounds of training, averaged over 5 independent runs. The number of rounds depends on the dataset and the model used, and it is equal to all the algorithms. For example, for the CIFAR-10 dataset and LeNet, the round budget is fixed at 10.000 rounds (for more details, see Table 8)
> We hope that our answer addresses your concerns and that our revision will contribute positively to the evaluation of our work. Please let us know if there are any remaining concerns and if our answers clarify your doubts.
>
> [1] Breaking the centralized barrier for cross-device federated learning, NeurIPS 2021 \
> [2] Federated Visual Classification with Real-World Data Distribution, ECCV 2020

---

> > ### Comment · Reviewer_wwM7 · 2023-11-23
> >
> > Many thanks for your answers which clarify the experiments, I upgraded my score accordingly.

---

### Author Response · Authors · 2023-11-16
**Global Message to Reviewers**

We thank all the reviewers for the time and effort dedicated to reviewing our work. We are glad that most of them appreciated:
* The **clarity of our writing**, which makes it dynamic (wwM7) and easy to follow (zlt2, Yr1w)
* Our **attention to an important problem in FL**, that is communication-efficiency under statistical heterogeneity (zlt2)
* Our analysis of both **cross-silo** and **cross-device** scenarios (wwM7, zlt2)
* The **simplicity and cleverness of our novel approach** (wwM7, zlt2), and our **extensive validation** (zlt2, 8a6j)

Nonetheless, we observed that certain reviews' summaries do not adequately convey the scope and contributions of our work. Consequently, we endeavor to rectify this discrepancy by presenting a succinct outline.

This work studies the problem of communication-efficient FL under extreme statistical heterogeneity; its main contributions are:
* **Demonstrating the limitations of SOTA FL algorithms:** Our findings reveal that current sota FL algorithms lack robustness when confronted with extreme statistical heterogeneity (fig 1, tab. 10) and large-scale scenarios (tab. 4).
* **A novel generalization of momentum:** We propose a Generalized Heavy-Ball (GHB) momentum formulation, which consists of calculating the momentum term over a window of $\tau > 1$ rounds. The guiding intuition is that, as $\tau$ increases, the momentum term progressively includes the information of more and more clients, and hence can provide more effective correction to client drift.
We demonstrated that having $\tau>1$ is crucial to developing the robustness against extreme heterogeneity we claim (see Fig. 3).
Conversely, we also show, both analytically and experimentally, that existing momentum-based FL algorithms are special cases of GHB with $\tau=1$, bringing some order to the literature of momentum in FL.
* **FedHBM method:** Within the GHB formulation, we developed communication-efficient declinations, which self-control the window length according to client participation. The optimality of this choice is motivated by the fact that, when equal to the length of the average period (e.g. $\tau = \frac{1}{C}$), it should contain the information on the global distribution. We experimentally verified this hypothesis, as we showed in Figures 2-3.
All the results converge to our claim: FedHBM is the most communication-efficient FL algorithm (table 2) and it yields the best accuracy (tables 3-4-5-10).
* **Extensive validation:** As our work is primarily experimental, we devoted much effort to performing extensive evaluation across computer vision and NLP tasks, multiple architectures (LeNet, ResNet-20, MobileNetv2), and large-scale real-world datasets. In fact, in our opinion experiments are the best testing ground to prove the robustness of algorithms, and hence their applicability in the real world.
In this light, we tested against extreme heterogeneity usually not proposed in other works, and extended the evaluation to cross-device settings, even if our algorithm is primarily intended for cross-silo FL. Yet, we proved remarkable versatility to such scenarios, corroborating the validity of our solution for practical deployment.
* **Reproducibility:** Throughout this work we strived to apply the best deep learning practices, providing a comprehensive description of the experimental setup and hyperparameter search for all the algorithms we tested (see section B.1, especially table 7). We are also attaching our code for FedHBM, to remark our commitment to full reproducibility.

We took into consideration your suggestions and updated the manuscript accordingly. For clarity, we put all the changes with respect to the submitted version in blue.

---

### Meta-Review · Area_Chair_x6zE · 2023-12-11

**Metareview:**

The paper considers the well-studied setting of federated learning with heterogeneous data. While using (server) momentum as a stabilization to compensate for client drift in this setting has become a classic technique by now, the use of heavy-ball momentum however is a new variant and topic of the present paper. Experiments are provided only in rather unrealistic settings (no partition of Cifar10 can mimic a realistic cross-device setting), and the achieved improvements could lie in the margin of different hyperparameter choices. Several reviewers also remarked that the hyperparameter search would have to be expanded and done separately per competing method, over more than just 3 values (Table 7 in the Appendix). Related work needs to be expanded (such as the 3 references mentioned by Reviewer Yr1w but not answered in the rebuttal). Learning rate schedules might have to be considered to be fair to competitors, as correctly mentioned by Reviewer 8a6j.

Overall the consensus unfortunately remained that this paper is rather straight-forward and not obviously a notable step compared to the literature. Reviewers also note that that no new theory or theoretical motivation was provided for the heavy-ball variant as compared to the classic momentum-based ones.

We hope the detailed feedback helps to strengthen the paper for a future occasion.

**Justification For Why Not Higher Score:**

Incremental novelty

**Justification For Why Not Lower Score:**

N/A

---

### Decision · Program_Chairs · 2024-01-16

Reject